# Label-Free Cancer Detection Methods Based on Biophysical Cell Phenotypes

**DOI:** 10.3390/bioengineering12101045

**Published:** 2025-09-28

**Authors:** Isabel Calejo, Ana Catarina Azevedo, Raquel L. Monteiro, Francisco Cruz, Raphaël F. Canadas

**Affiliations:** 1Department of Biomedicine, Faculty of Medicine, University of Porto, 4200-450 Porto, Portugal; 2RISE-Health, Faculty of Medicine, University of Porto, 4200-450 Porto, Portugal; 3ESS—Escola Superior de Saúde, Instituto Politécnico do Porto, 4200-072 Porto, Portugal; 4Urology Department, Unidade Local de Saúde de São João, 4200-319 Porto, Portugal; 5Department of Surgery and Physiology, Faculty of Medicine, University of Porto, 4200-450 Porto, Portugal

**Keywords:** biophysical biomarkers, cancer cells, label-free methods, cell sorting, cancer diagnostics

## Abstract

Progress in clinical diagnosis increasingly relies on innovative technologies and advanced disease biomarker detection methods. While cell labeling remains a well-established technique, label-free approaches offer significant advantages, including reduced workload, minimal sample damage, cost-effectiveness, and simplified chip integration. These approaches focus on the morpho-biophysical properties of cells, eliminating the need for labeling and thus reducing false results while enhancing data reliability and reproducibility. Current label-free methods span conventional and advanced technologies, including phase-contrast microscopy, holographic microscopy, varied cytometries, microfluidics, dynamic light scattering, atomic force microscopy, and electrical impedance spectroscopy. Their integration with artificial intelligence further enhances their utility, enabling rapid, non-invasive cell identification, dynamic cellular interaction monitoring, and electro-mechanical and morphological cue analysis, making them particularly valuable for cancer diagnostics, monitoring, and prognosis. This review compiles recent label-free cancer cell detection developments within clinical and biotechnological laboratory contexts, emphasizing biophysical alterations pertinent to liquid biopsy applications. It highlights interdisciplinary innovations that allow the characterization and potential identification of cancer cells without labeling. Furthermore, a comparative analysis addresses throughput, resolution, and detection capabilities, thereby guiding their effective deployment in biomedical research and clinical oncology settings.

## 1. Introduction

Cancer, marked by abnormal and uncontrollable cell growth, encompasses a wide range of pathologies originating from almost any organ or tissue [1]. As the second leading cause of mortality globally [1], cancer’s poor prognosis and limited survival rates are often linked to late-stage diagnosis and advanced disease progression [1,2]. Conventional diagnostic methods, such as magnetic resonance imaging, biopsy histology, tomography, and invasive imaging (e.g., cystoscopy) [3], although effective, remain expensive and require sophisticated equipment and specialized clinicians. Moreover, these methods can be limited in resource-constrained settings and have inherent invasiveness [4,5]. Therefore, it is imperative to develop complementary and alternative non-invasive methodologies that are cost-efficient and easily automated for cancer diagnosis and prognosis. In this context, biophysical biomarkers such as extracellular matrix stiffness and cellular mechanical cues have gained interest as potential indicators for detecting cancer. These biomarkers influence key cellular processes such as differentiation, growth, and the motility of tumor cells [6,7]. For the identification and analysis of biophysical changes, advanced technologies, which will be mentioned later, are successfully used for macroscopic tumor evaluation.

Particularly, label-free methods stand out for their ability to detect unique cellular features without the need for labeling agents. These methods offer lower workload, minimal sample damage, cost-efficiency, and ease of integration into lab-on-a-chip and benchtop systems [8]. By focusing on the biophysical characteristics of cells, label-free methods provide more accurate and reproducible results when compared to label-dependent methods [8,9]. Currently, label-free technologies encompass both conventional methodologies and cutting-edge innovations, including phase-contrast microscopy, which may be powered by AI-integrated algorithms, holographic microscopy, varied cytometric analysis, microfluidics, and adaptations from other disciplines like dynamic light scattering (DLS), atomic force microscopy (AFM), or electrical impedance spectroscopy (EIS) [10,11,12,13]. These techniques allow a fast [12] and label-free cell identification, dynamic monitoring of cellular interactions [10], visualization of spatial constructions [11] and morphological cues, with a strong focus on the biophysical properties of cells, such as mechanical, magnetic, and electrical features [13]. Such capabilities are highly valuable in several biomedical applications, particularly in liquid biopsies, where circulating tumor cells (CTCs) are found to act as minimally invasive sources of information for the diagnosis and evolution of cancer [10,11,12,13]. CTCs, consisting of cancer cells released into the bloodstream from primary tumor lesions [14], contribute to metastasis’ origin, therefore being pointed out for early detection of metastatic cancer [15].

In this regard, this review provides an integrated perspective on the current landscape of label-free technologies, with an emphasis on their application in cancer detection and monitoring through cellular biophysical features, particularly in the context of liquid biopsies (e.g., urine, blood). Nonetheless, advantages, challenges, and clinical translational potential to detect biophysical biomarkers are also discussed, demonstrating the potential of these bio-tools in the evolving field of cancer diagnostics.

## 2. Microfluidics Systems for Cell Separation and Enrichment

Microfluidics involves the manipulation and analysis of small fluid volumes within microscale channels, typically measuring hundreds of micrometers in width [16]. Operating at the submillimeter scale, these systems offer several advantages, including high sensitivity, minimal reagent consumption, cost-effectiveness, and enhanced spatiotemporal resolution [17]. Despite these benefits, challenges remain, particularly in scalability, due to complex device design, prototyping, and fabrication. Other limitations include sample preparation, channel clogging, and surface fouling from biological materials, all of which can compromise reliability and limit further implementation in clinical settings [18]. Nevertheless, the integration of high-throughput microfluidic platforms in oncology is revolutionizing the research field. A notable example is a system capable of generating approximately 12,000 tumor spheroids per chip [19]. This platform allows drug screening in a 3D environment that closely mirrors in vivo tumor conditions, offering a clear improvement over traditional 2D cultures. Importantly, cell viability assessments conducted in a non-destructive, label-free manner agreed with conventional viability assays, demonstrating compatibility with numerous compounds and cell lines [19]. Such innovations hold significant promise, not only for advancing drug development but also for personalized treatment strategies. By offering patient-specific cancer models, these microfluidic platforms have the potential to predict therapeutic responses and assess disease prognosis without the need for labeling or invasive procedures. The following sections delve into each of these microfluidic strategies, exploring their operating principles, recent developments, and relevance for precision oncology, particularly based on label-free approaches and cellular biophysical biomarkers.

### 2.1. Passive Cell Separation Microfluidics

Passive microfluidics refers to the manipulation of flows in micro-structured devices that operate without external forces, such as electric, magnetic, or acoustic fields [20]. By relying solely on the intrinsic hydrodynamic behavior of particles, passive systems are simpler to operate and generally more reliable. Importantly, passive, label-free microfluidic devices can achieve rapid cell separation with high-throughput compared to active microfluidic systems [20].

#### 2.1.1. Deterministic Lateral Displacement

Deterministic lateral displacement (DLD) is a widely used passive method that segregates suspended particles based on their structural dimensions. Its principle relies on directing particles through a microchannel array of regularly spaced pillars, which causes particles above a certain critical diameter to be deflected along specific trajectories [20]. Larger particles collide with the pillars and follow a displacement mode along the array, while smaller particles follow a zigzag pattern, remaining largely in the fluid stream [20,21,22].

Advances in DLD chip design have significantly enhanced separation efficiency [20]. One such application involves a dual-stage microfluidic biochip engineered for isolating CTCs and tumor clusters (Figure 1A). In this system, the first stage uses a DLD array in the upper microchannels to enrich tumor clusters in a label-free manner, achieving 88.6% recovery and 92.2% purity. A secondary antibody-coated lateral filter then captured individual tumor cells and smaller clusters missed in the first stage, achieving 89.54% recovery and 89.44% purity [23]. Although not entirely label-free, the combination of passive physical separation and affinity-based selection illustrates the potential of hybrid microfluidic systems to maximize cell yield and purity.

A critical factor in DLD systems’ performance is the pillar design. Poorly optimized pillar shapes can destabilize zigzag trajectories for smaller particles, reducing separation accuracy. To address this, topology-optimized DLD chips have been developed with refined pillar shapes and channel widths [20], achieving miniaturization, improved resistance to clogging and tighter control over the critical diameter. Validation with polystyrene beads and cancer cells demonstrated better capture purities exceeding 92.5% and recovery rates of 97.1%, outperforming conventional DLD designs (Figure 1B) [20].

Altogether, DLD offers a robust, efficient label-free approach for the separation of tumor cells based on structural dimensions. The continued advancements in chip design, like geometry optimization and integration with secondary capture strategies such as immune affinity, are significantly expanding DLD’s applicability in cancer diagnostics. Nonetheless, current DLD platforms remain limited to size-based discrimination, with minimal sensitivity to cellular deformability or shape. Therefore, integrating DLD with complementary active sorting or functional assays would improve specificity and enable functional characterization, further increasing translational value.

#### 2.1.2. Inertial Focusing and Centrifugal Microfluidics

Inertial focusing and centrifugal microfluidics leverage hydrodynamic and rotational forces to passively manipulate and separate cells or particles within microchannels, respectively [24]. These platforms guide particles to equilibrium positions determined by their size and density and influenced by factors such as channel geometry and flow conditions [24]. It provides a high-throughput label-free method useful for point-of-care diagnostics.

Recent advances have demonstrated the potential of inertial focusing for isolating rare tumor cells from urine samples using spiral microchannels [25]. By exploiting a combination of inertial lift and Dean drag forces, cells migrate laterally toward a size-dependent equilibrium position, where larger cells accumulate near the inner wall, while smaller ones disperse outward. This technique achieved a separation efficiency of ~85%, with results strongly correlating with traditional diagnostic markers like prostate-specific antigen (PSA) levels and Gleason scores [25]. These findings highlight the method’s promise as a non-invasive prostate cancer screening tool (Figure 1C). In another application, a cost-effective centrifugal microfluidic platform was developed to isolate CTCs from whole blood. This platform used a Y-shaped microchannel integrated with a contraction–expansion array (CEA) and a bi-furcation region, inducing differential flow velocities that separated cancer cells from blood components with up to 90% efficiency [26]. These results demonstrate the clinical utility of inertial platforms for early cancer detection via liquid biopsy, aligning with key prognostic biomarkers.

One of the standout devices in this field is the CTC-iChip, which combines passive inertial focusing with active magnetic separation. Developed by Ozkumur et al. [27], this hybrid system captured both EpCAM-positive and EpCAM-negative cancer cells from blood, overcoming the limitations of traditional affinity-based CTC capture methods [27]. With a processing rate of 10 million cells per second, the CTC-iChip enabled downstream morphological, immunohistochemical, and molecular analyses at the single-cell level. It has been successfully applied to multiple cancer types, including lungs, prostate, pancreatic, breast, and melanoma, providing a comprehensive tool for tumor profiling and monitoring disease heterogeneity.

Altogether, inertial and centrifugal microfluidic methods, using physical properties like size and shape, offer a scalable, simple solution for clinical use. While these systems are advantageous in throughput and simplicity, they may struggle to separate cell types with similar biophysical features. Combining these with active techniques, like magnetic separation, will enhance separation efficiency and enable the capture of a broader range of cells. This hybrid approach holds great promise for more precise, comprehensive diagnostics, particularly in cancer.

#### 2.1.3. Microfiltration

Microfiltration is a passive cell separation technique that uses size-based filtration to selectively extract specific cell populations from heterogeneous mixtures [21]. This method employs microfilters with precisely dimensioned microstructures that capture cells exceeding a predetermined size limit, while allowing smaller ones to pass through. In addition to size, other physical properties, such as cell morphology and rigidity, can also be exploited to enhance selective separation [21]. Unlike DLD, which relies on flow dynamics and inertial forces, microfiltration relies solely on physical barriers.

To overcome challenges in traditional microfluidic technologies, such as non-scalable fabrication, processing times, and a lack of automation, advanced membrane-based microfiltration devices have been developed. One such device integrates fully automated sample processing with machine-vision imaging, improving the efficiency of CTC isolation from blood (Figure 1D) [28]. Using nickel membranes, this device achieved greater than 93% capture efficiency for several prostate cancer cell lines (including PC-3, VCaP, DU-145, and LNCaP), with high reproducibility, indicated by low coefficients of variation (<7%) [28]. It processed 7.5 mL blood samples in under 12 min, with captured tumor cells showing high viability (averaging 90.3%), minimizing damage from shear stress during filtration. Moreover, this system supported downstream molecular analyses like mutation detection, even at low CTC concentrations, making it valuable for clinical monitoring and disease management [28].

Moreover, the microfiltration technique has been applied for the negative enrichment of CTCs from whole blood, selectively depleting normal blood cells while preserving tumor cells [29]. This strategy reduced bias and increased the technology’s applicability across different tumor types. Notably, Chia-Heng Chu et al. [29] developed a 3D-printed microfluidic device that combined leucodepletion channels with microfiltration. It effectively separated tumor cells from blood based on size contrast with normal cells, achieving a 2.34-log depletion by capturing over 99.5% of white blood cells from 10 mL of whole blood while recovering more than 90% of tumor cells [29]. This device successfully isolated CTCs from prostate and pancreatic cancer patient samples, with measured concentrations ranging from 0 to 3.4 CTCs/mL in prostate cancer samples and around 0.3 CTCs/mL in pancreatic cancer samples [29]. While these concentrations are relatively low, highlighting the challenge of isolating rare CTCs, they still demonstrate the device’s significant potential for non-invasive tumor diagnostics and liquid biopsies based on biophysical features of cells.

In conclusion, microfiltration is a powerful and versatile tool for cell separation, offering significant improvements in isolation efficiency, throughput, automation, and compatibility with downstream analyses. However, as highlighted, the low concentration of CTCs in certain samples may present challenges in detecting and isolating tumor cells, which warrants further optimization for more sensitive applications. Combining microfiltration with active separation methods further enhances accuracy. Technique’s adaptability to various clinical oncological applications underscores its promise in enhancing diagnostic capabilities and personalized medicine.
Figure 1Passive microfluidic platforms for cell biophysical separation and enrichment. (**A**) Size- and affinity-based tumor cluster isolation using a two-layer microfluidic biochip. (i) Top view schematic of the chip design, featuring a DLD module that isolates large tumor clusters based on size, and a lateral filter section that captures smaller clusters and individual tumor cells via size and immunoaffinity. (ii) THP-1 cells were effectively washed out, while the number of captured 4T1 cells varied depending on the reverse PBS buffer flow rate, as shown in the accompanying graphs. Qualitative comparison images below show captured cells before and after purification. Scale bar: 100 µm. Reproduced with permission from [23]. (**B**) Cell sorting performance in a microfluidic array. (a) Size comparison between blood cells and ECA-109 (human esophageal squamous cell carcinoma) cells. (b) Quantification of mean lateral displacement and outlet distribution for RBCs, WBCs, and ECA-109 cells. (c) Outlet distribution comparison between a topology optimized array (top) and a conventional circular array (bottom), showing improved separation efficiency. Reproduced with permission from [20]. (**C**) Spiral microfluidic chip for PCa cell isolation from urine. (i) Schematic of sample processing through a spiral microchannel, including a magnified view of the bifurcation zone. (ii) At an optimal flow rate of 1.7 mL/min, approximately 85 ± 6% of PCa cells were efficiently collected at the designated outlet. Reproduced with permission from [25]; (**D**) Application of microfiltration-based CTC isolation. (i) Overview of the microfiltration device used for CTC isolation from blood. (ii) CTC counts from blood samples of 8 healthy male donors and 8 metastatic prostate cancer patients revealed significantly elevated CTC numbers in the patient group. Reproduced with permission from [28].
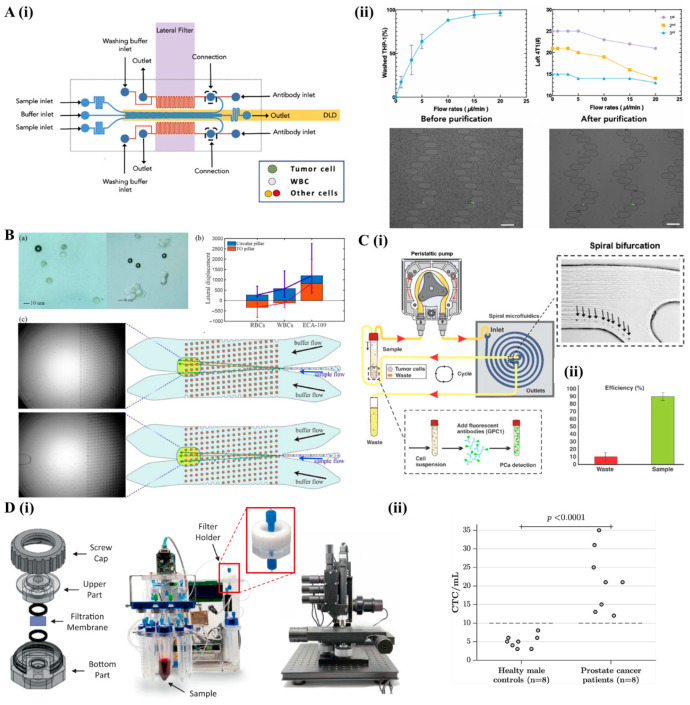



### 2.2. Active Cell Separation Microfluidic

Compared to passive systems, active microfluidic platforms offer more precision and tunability by applying external force fields to manipulate cells based on their unique biophysical properties. These platforms rely on physical forces, such as acoustic, magnetic, and electric fields, to induce differential movement or positioning of cells, enabling precise separation of phenotypically similar or rare cell populations [30]. It makes them particularly valuable for applications like CTC isolation, tumor heterogeneity assessment and cell state profiling (Figure 2). The subsections below explore the main active separation methods—acoustofluidics, magnetofluidics, and dielectrophoresis—examining their principles, advantages, and emerging applications in label-free cancer cell detection.

#### 2.2.1. Acoustofluidics

Acoustofluidics, also known as acoustic microfluidics or acoustophoresis, has gained significant attention as a label-free technique for the manipulation and separation of cells within a microfluidic device [31]. By employing low-intensity acoustic pressure, specifically bulk acoustic waves (BAWs) or surface acoustic waves (SAWs), acoustofluidics separates cells based on their size, density, compressibility, and other physical properties. This method offers a non-contact, biocompatible, and continuous approach to sample processing, making it an attractive alternative for cancer cell isolation [31].

Traditionally, centrifugation using density gradients is commonly used for isolating blood components, like RBCs, WBCs, and platelets. However, this method has limitations. Density gradient centrifugation yields only 25–50% in efficiency and suffers from low biocompatibility, as it requires additional buffer preparations and can lead to exosome fusion and soluble protein contamination, which compromises the integrity of isolated particles [32]. In contrast, acoustofluidics provides a more efficient (~82% yield) biocompatible alternative by allowing continuous, gentle separation of cells without the need for chemical labels or extensive sample preparation. For example, in plasma exchange applications, acoustofluidics successfully guided RBCs into clean medium channels, achieving >95% recovery and 98% contaminant removal at a flow rate of 0.17 mL/min, while preserving cell viability during platelet and WBCs separation [32]. In oncology, acoustofluidics has proven particularly useful for isolating CTCs from blood—an inherently difficult task due to their low abundance. Using a tilted angle standing surface acoustic wave (SSAW) platform, CTCs were isolated at a flow rate of 1.2 mL/h with a recovery rate above 83%, even when samples contained as few as 100 CTCs/mL [33]. Optimization through finite element modeling of device geometry, tilt angles, and operational parameters ensures efficient cell focusing and separation [33].

A critical challenge in acoustofluidics is cell misalignment upon entry into the acoustic field, which can impact separation efficiency. To overcome this, Augustsson et al. [34] developed a pre-alignment system within a temperature-stabilized microchannel for prostate cancer cells detection. This improvement led to >99% purity when separating microspheres from 5 µm to 7 µm and achieved 93.6–97.9% recovery of fixed tumor cells (and 72.5–93.9% for viable cells) with purity reaching up to 99.7% [34]. These results underscore the need for pre-alignment to enhance separation fidelity for both viable and fixed cancer cells [34].

Another challenge in CTC analysis is the loss of rare cells during post-separation processing. Acoustofluidics has been effectively combined with dielectrophoresis (DEP) to further improve cancer cell handling efficiency (Figure 2A) [35]. In one study, acoustic waves were used to concentrate prostate cancer cells, which were then captured in electroactive micro-well arrays for direct analysis, minimizing the need for post-processing and reducing the risk of losing rare cells [35]. This way, the integrated platform allowed single-cell resolution analysis with high specificity, proving its potential in clinical applications.

Thus, acoustofluidics offers other advantages over traditional methods, due to its label-free nature, high scalability, and minimal sample disruption. It preserves cellular integrity, which is crucial for liquid biopsy applications involving rare cancer cells, such as those found in blood or urine samples. Combining acoustofluidics with methods like dielectrophoresis improves accuracy and further reduces cell losses with preserved viability. Future work focusing on device optimization and hybrid integration will strengthen its role in minimally invasive cancer monitoring and personalized medicine.

#### 2.2.2. Magnetofluidics

Magnetic microfluidics, or magnetofluidics, uses magnetic fields to manipulate cells, particles and fluids within microchannels [36]. While magnetic forces are often used in label-dependent applications (e.g., with magnetically tagged particles), magnetofluidics can also be implemented in label-free modes by exploiting intrinsic biophysical properties such as mass density and magnetic susceptibility [36].

A key example of label-free magnetofluidics is magnetic levitation, which enables real-time monitoring and characterization of cells based on their density differences. This technique uses a paramagnetic medium, often gadolinium-based, suspended between two magnets to levitate cells at equilibrium heights determined by their biophysical properties [37]. Magnetic levitation allows real-time characterization with a cell mass density resolution of 1 × 10^−4^ g·mL^−1^, which is 10 to 100 times finer than density gradient centrifugation, and at least 10 times higher in resolution compared to passive microfluidic sorting methods [37]. Moreover, this method has allowed single-cell density quantification of various cancer types, including breast (1.044 g/mL), esophageal (1.059 g/mL), and lung cancer cells (1.062 g/mL), demonstrating its applicability in cancer cell phenotyping [37].

Another prominent example of magnetofluidics is magnetophoresis, where magnetic fields are used to control and separate ferrofluids based on their magnetic properties (Figure 2B). It has successfully been used to separate cancer cells from leukocytes, achieving an average separation efficiency of 82.2% with a yield of 1.2 mL/h [38]. In particular, the technique was able to isolate cancer cells at very low concentrations (~100 cells/mL) from leukocyte samples (~10^6^ cells/mL), maintaining a post-separation viability of 94.4% and demonstrating the biocompatibility of ferrofluid systems [38]. Moreover, the cytocompatibility of ferrofluids was confirmed, as cancer cells not only survived exposure but also proliferated normally post-detachment [38].

In a different study, magnetofluidics has also shown promise in distinguishing cancer cells with varying invasive potentials. Magnetic levitation was applied to sort invasive and non-invasive breast cancer cells (MCF-7 and MDA-MB-231, respectively) from monocytic U937 cells (isolated from a patient with histiocytic lymphoma) in a label-free manner [39]. Despite minimal density differences (~0.011 g/mL), the system achieved sorting efficiencies of approximately 70%, while processing samples at a rate of 1 mL/h [39]. The results underscore the technique’s sensitivity in discriminating cell types with overlapping physical traits while preserving cell integrity for downstream analyses. Notably, the separation was achieved at an even higher rate than the previously reported 1.2 mL/h.

Beyond CTCs, circulating hybrid cells (CHCs) have gained attention in recent magnetofluidics studies due to their unique characteristics, combining attributes of both macrophages and neoplastic cells, and their functional role in metastatic spread [40]. These hybrid cells, often more abundant than conventional CTCs in cancer patients [41], have been found to levitate and focus on different heights under paramagnetic conditions, allowing quantification of their biophysical properties. Specifically, CHCs exhibited levitation heights of 320 ± 29 μm in a 30 mM medium, which increased to 422 ± 19 μm and 468 ± 27 μm in higher concentrations of paramagnetic medium of 50 nM and 80 mM, respectively, contrasting with the lower levitation heights of peripheral blood mononuclear cells (PBMCs) (average of 226 ± 58 μm in 30 mM, 295 ± 75 μm in 50 mM, and 400 ± 84 μm in 80 mM) [40]. These findings highlight the distinct biophysical properties of CHCs, which can be exploited for their detection and isolation among healthy cells.

In sum, magnetofluidics is a powerful and versatile platform for cancer cell separation and detection, leveraging intrinsic magnetic susceptibilities to isolate rare cells with high efficiency and minimal labeling. Compared to traditional methods like density-based centrifugation or flow cytometry, magnetofluidics systems offer label-free sorting specificity, particularly in capturing CTCs based on their inherent single-cell density lower levitation heights. In addition, it allows detailed cell mass biodensity characterization. While magnetofluidics offers promising capabilities, the need for complex and expensive apparatus for micromachining remains a significant obstacle to the rapid and low-cost fabrication of lab-on-chip (LOC) systems based on magnetophoresis. This limitation can hinder the widespread accessibility and scalability of these technologies for broader applications [42]. Nevertheless, advances in ferrofluid formulations, levitation chamber designs, and hybrid technologies will likely elevate its clinical relevance in non-invasive cancer diagnostics and personalized oncology.

#### 2.2.3. Dielectrophoresis Microfluidics

Dielectrophoresis (DEP) is an electrokinetic phenomenon widely used in microfluidic platforms for manipulating cell movement [43]. It uses non-uniform electric fields, leveraging differences in cells’ dielectric properties, such as membrane capacitance, cytoplasmic conductivity, and overall polarizability [43]. Briefly, when subjected to an electric field, cells experience a force depending on their size, shape, and intrinsic electrical properties that allows label-free separation between different cell types or functional states.

DEP platforms operate in two primary modes: positive DEP (pDEP), where cells are attracted to regions of high electric field intensity, and negative DEP (nDEP), where cells are repelled from these regions [44]. The response of cells to these forces is frequency-dependent, varying significantly between cell types, which makes DEP a flexible and tunable method for biophysical cell sorting.

In oncology research, DEP has proven effective for isolating CTCs from leukocytes without compromising cell viability [45]. In a study evaluating different lateral electrode configurations, a semicircular electrode arrangement was found to be the most effective, achieving a recovery rate of nearly 95% for MDA-MB-231 breast cancer cells. The semicircular configuration applied a maximum electric field of 1.11 × 10^5^ V/m, below the threshold required for cell electroporation, ensuring that the cells maintained their integrity during the separation process. Joule heating studies also indicated minimal temperature variation, approximately 1 K, which is a safe range for maintaining cell viability within the microchannel fluid. Overall, the findings confirm the effectiveness of dielectrophoresis in tumor cell separation [45].

In the context of CTC isolation from blood samples, including red blood cells (RBCs), white blood cells (WBCs), and platelets, a microfluidic system incorporating front electrode configuration dielectrophoresis (FEC-DEP) has demonstrated significant promise. (Figure 2C). Numerical simulations were employed to optimize system parameters for improved separation efficiency and purity, resulting in an approximately 80% separation rate for CTCs. This highlights the technique’s potential for non-invasive tumor cell isolation from blood samples. However, the alignment and positioning of electrodes can sometimes introduce inconsistencies in cell capture, which can impact reproducibility. These factors can pose challenges in scaling up the technique for high-throughput applications or in clinical settings where consistent results are critical for proof-of-concept [46].

Moreover, DEP has shown promise in the capture and detection of other biomarkers of tumors: small extracellular vesicles (sEVs), particularly found in urine. A specific DEP, known as the acDEP-Exo chip, was developed to detect sEVs with a sensitivity limit of 161 particles/µL, far surpassing conventional methods such as immunoaffinity and ultracentrifugation [47]. In another study using plasma samples from breast cancer patients and healthy volunteers, the combination of biomarkers EpCAM and MUC1 provided high diagnostic accuracy, showcasing DEP’s potential for non-invasive biomarker detection without the need for immunofluorescence methods, which can suffer from contamination and mechanical damage issues [47].
Figure 2Active microfluidic platforms for cell separation using acoustic, magnetic, and electric fields. (**A**) Acoustophoresis and dielectrophoresis-based cell manipulation. (i) Schematic of the integrated microfluidic device, highlighting the functional zones of the acoustophoresis chip for size-based separation and the DEP trapping array for cell capture, with dimensions indicated. (ii) Time-lapse imaging shows the trapping zone (white arrow) of separated cancer cell clusters (orange) by the DEP array. Scale bars: 100 µm. Reproduced with permission from [35]. (**B**) Ferrofluid-based biocompatible cell separation. (i) Schematic of the microfluidic separation process where the cell sample, ferrofluid, and buffer are injected without pre-mixing. During operation, cells are exposed to ferrofluids only while being actively separated. Larger cancer cells are deflected into the buffer stream for collection, eliminating the need for additional washing. (ii) Left: Experimental image showing A549 cancer cells reaching the ferrofluid/buffer interface and exiting through the outlet. Right: Fluorescence microscopy image of A549 cells stained with CellTracker Green during sorting; dashed white lines indicate microchannel boundaries. Scale bars: 200 µm. Reproduced with permission from [38]. (**C**) Dielectrophoresis-based CTC isolation. (i) Design of a microfluidic chip incorporating a FEC-DEP for dielectrophoretic separation of CTCs from blood. (ii) Graph illustrating the influence of the velocity ratio between the buffer and sample inlets on overall separation efficiency. Reproduced with permission from [46].
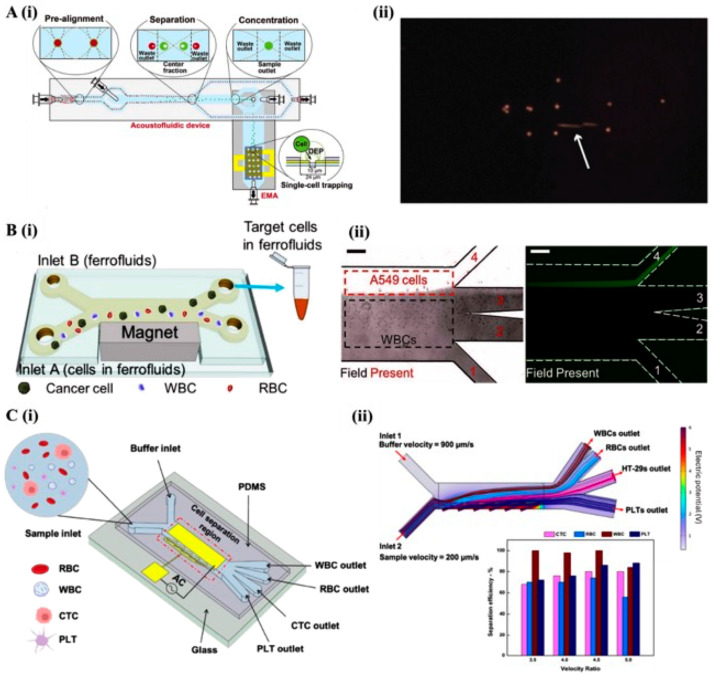



Thus, dielectrophoresis stands out as a precise technique for the manipulation and separation of cancer cells. The use of optimized semicircular and frontal electrode configurations demonstrated high efficacy in the recovery and preservation of tumor cells, ensuring their viability. However, there is still a challenging need to fine-tune parameters for each specific cell type and sample matrix. Proper calibration and alignment of electrode configuration are essential to minimize issues and ensure reproducible performance. Furthermore, the technique has shown promise in detecting tumor biomarkers by capturing sEVs, which are predominantly found in urine, thereby enhancing non-invasive diagnostic approaches. These advances position dielectrophoresis as a valuable method for cell separation and non-invasive diagnosis of diseases like cancer, contributing significantly to the development of high-precision diagnostics.

### 2.3. Cellular Mobility Within Microfluidic Arrays

Microfluidics designed for cell separation based on their path have also been developed to address challenges in isolating metastatic tumor cells, often present in low quantities in mixed body fluids. One such platform (Figure 3) focuses on separating and enriching cancer cells by exploiting their biomechanical properties [48]. Cancer cells typically have different biomechanical properties, such as size and stiffness, compared to healthy cells, allowing them to be directed to other outputs of the device. Larger, stiffer cells tend to move toward specific outlets, while smaller, softer cells pass through constrictions and are collected at different exits. Cell path analysis has shown that cells with greater metastatic potential tend to be softer and display different migratory behaviors compared to less malignant cells, allowing for effective separation [48].
Figure 3Trajectory-based cancer cell enrichment microfluidic platform. (**A**) The device uses 3 inlets and 5 outlets to sort cells. Sheath flow focuses cells to the ridge edge, where angled constrictions force them to deform. Stiff, large cells are deflected toward outlets 1–3, while soft, small cells pass under the ridges to outlets 4–5 (scale bar = 3 mm). (**B**) (i) Representative movement patterns of a metastatic ovarian cancer cell line (HEY) compared to a non-metastatic line (IOSE); (ii) Migration assay comparing the invasiveness of HEY and OVCAR-3 cells; (iii) Analysis of cell trajectories across ridges for various ovarian cancer cell lines, showing that cells with lower metastatic potential (IOSE, OVCAR-3) experienced greater deflection than highly metastatic cells (HEY, HEY-A8) (Reproduced, with permission, from [48]).
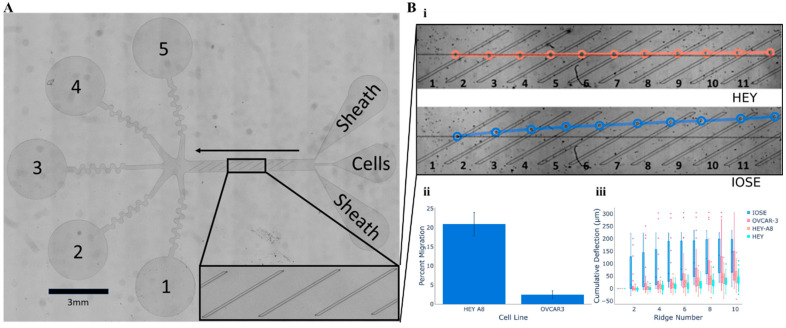



Other microfluidic designs have become relevant for studying cell migration in 3D environments [49]. Metastasis, responsible for over 90% of cancer-related deaths, relies on the complex migration of tumor cells within the tumor microenvironment, including the extracellular matrix. Unlike conventional 2D systems, 3D microfluidic models replicate the biomechanical and biochemical properties of the tumor microenvironment, offering a more accurate representation of cell migration and invasion [49]. Recent studies revealed that cell migration is influenced by factors such as matrix stiffness, interstitial flow and interactions between tumor cells and cancer-associated fibroblasts. Furthermore, tumor cells exhibited distinct migratory behaviors, alternating between individual and collective migration, in response to biochemical and biomechanical signals [49]. Real-time visualization of cellular dynamics allows for the observation of these migratory behaviors under different experimental conditions, providing valuable insights into the metastatic process. To illustrate this, a study revealed that metastatic HEY and OVCAR-3 ovarian cancer cells exhibited less deflection when passing through microfluidic ridges compared to IOSE non-metastatic cell lines, which were more strongly deflected [48]. This finding highlights the differences in biomechanical properties that correlate with metastatic potential (Figure 3).

Cell migration was also assessed using microfluidics integrated with the electrical cell–substrate impedance sensing (ECIS) technique, allowing real-time monitoring at the single-cell level. ECIS methodology shows effectiveness in monitoring cellular events such as adhesion, growth, and motility, through electrical changes at the interfaces between cells and electrodes [50]. Thus, it verified significant changes in the magnitude of impedance in metastatic cells and allowed cellular analysis of individual cells. This way, the combination of microfluidics and ECIS represents a fast and selective detection of the migratory properties of cancer cells at the single-cell level [31,50].

## 3. Label-Free Microscopy Techniques for Cellular Analysis

### 3.1. Phase-Contrast Microscopy—Morphological Profiling and Segmentation

Phase-contrast microscopy is a useful label-free imaging technique for observing transparent or reflective biological samples. This method exploits the phase shift in light waves passing through the sample, converting these shifts into visible contrast to enhance the image [51]. By amplifying interference patterns, phase-contrast microscopy provides high-contrast images that reveal fine details of the sample, while examining living cells in their natural state, without staining [52]. This allows researchers to observe and record the dynamic processes of living cells with exceptional clarity and resolution [52]. However, one limitation of this technique is that it cannot reliably provide quantitative information about the optical or physical thickness of the observed structures unless the optical thickness is significantly reduced [53].

To overcome this shortcoming, a deep cancer cell detector based on the Faster region-based convolutional neural networks (R-CNN) framework coupled with a Circle Scanning Algorithm (CSA) for detecting adherent cells was proposed by Zhang et al. [54] The system integrates region proposal generation with deep learning for object identification. By combining it with a specialized algorithm, the framework significantly improves the detection of individual cells in phase-contrast images, even at low cell densities. Moreover, a comparative analysis revealed that it outperformed previous hybrid methods by 19%, highlighting phase-contrast potential in cancer cell identification [54].

Further research using phase-contrast imaging has allowed for detailed morphological analysis of single cells, revealing distinct phenotypic signatures and functional diversity within different cell lineages (Figure 4A) [55]. By applying a deep learning algorithm for cell segmentation at varying densities, the technique enables precise characterization of cellular properties, including size, texture, and shape [56]. Furthermore, it allowed the study of cellular morphology diversity, allowing a detailed classification of different cell types and confluence stages. Analysis of these images showed significant variation in cell size, with some cells exceeding 6000 µm^2^, and the number of cells per image ranging from just a few to over 3000 [56].

Jo et al. introduced an innovative method for cells’ precise segmentation and tracking in phase-contrast microscopic images [57]. Although it did not directly target cancer detection, it was instrumental in studying cell migration, a key aspect of metastasis. The proposed method demonstrated high accuracy in tracking cancer cell migration, offering valuable insights into the mobility and behavior of cancer cells [57]. Thus, the methodology surpassed manual tracking techniques, improving temporal efficiency and providing precise results using simple but robust algorithms that allowed for automated, high-precision execution [57].

In another study, a hybrid approach combining phase-contrast microscopy and a resonant wave grating (RWG) biosensor for cell adhesion kinetics was developed [58]. This method allowed the segmentation and classification of cancer cells with high precision (Figure 4B), reaching the highest rate on fibronectin-functionalized surfaces. Furthermore, phase-contrast microscopy data integration with cell adhesion measurements provided a detailed overview of cellular activity. Testing across seven cell types, six cancer types (including breast and lung cancer) and one healthy cell type demonstrated that this approach could be applied to real-world automatic diagnostics. Using over 12,000 cell samples and employing various analysis techniques (e.g., F1-Score, AUC Score, and AUC-PR Score), the best models achieved nearly 95% classification accuracy. Combining these techniques not only improved the analysis of individual cells but also deepened the understanding of cell adhesion mechanisms, critical for cancer progression [58].
Figure 4Phase-contrast microscopy images and computational cell segmentation. (**A**) Image processing and morphological analysis. (i) Instance segmentation is used to extract individual cell masks, from which morphological features such as area, circularity, and aspect ratio are computed (second panel from top). A cell network is then constructed by linking neighboring cells based on spatial proximity, where each cell is a node, and edges represent adjacency (second from bottom). This network can be visualized as an abstract topology graph, independent of spatial coordinates (bottom). (ii) Uniform Manifold Approximation and Projection (UMAP) clustering of YM and YMR cell populations based on network-derived morphological features. Subclass distributions reveal distinct phenotypic groupings, highlighting representative single-cell clone (SCC) lineages selected for downstream 3D analysis. Color gradient denotes local cell density, ranging from 0 to 5 × 10^−4^ cells/µm^2^, emphasizing regions of phenotypic enrichment across the projected feature space. Reproduced with permission from [55]. (**B**) AI-assisted label-free optical imaging for single-cell detection and segmentation. (i) Aligned projection of the adhesion image indicating centroids of single-cell segments (blue markers). Scale bar: 150 µm. (ii) Four segmentation strategies are illustrated for selected cells: red (aligned microscopy segment), green (maximum pixel), blue (cover pixel-based), and yellow (watershed-based). Cell pose predictions (P) are also included. Columns C and D show a challenging region with overlapping MCF-7 cells, highlighting segmentation robustness in complex cases. Scale bars: 50 µm. Reproduced with permission from [58].
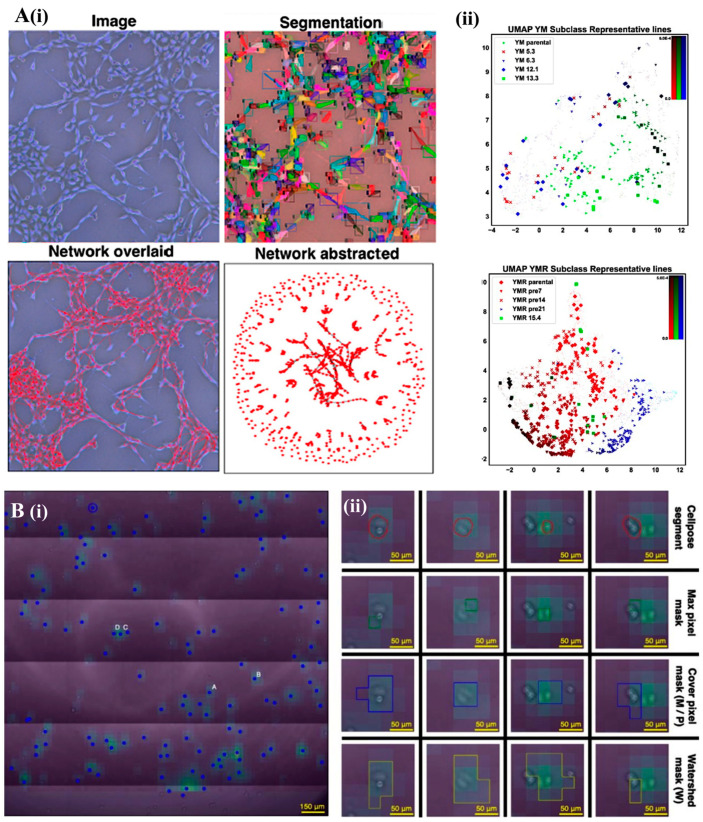



Phase-contrast microscopy has significantly evolved, particularly when combined with complementary techniques, and has proven to be a highly effective method for the detailed observation and analysis of cancer cells. However, challenges remain, including the limitations in quantifying optical thickness, difficulties in segmenting densely packed or overlapping cells, and the potential for segmentation errors in complex environments. Additionally, the need for significant computational resources and manual validation of automated results can be time-consuming. Despite these challenges, the combination of phase-contrast microscopy with advanced analytical approaches continues to offer promising applications for cancer research and beyond.

### 3.2. Holographic Microscopy: Quantitative Phase Imaging for 3D Characterization

Holographic microscopy (Figure 5A), or quantitative phase microscopy (QPM), is a non-invasive imaging technique that allows 3D label-free visualization of cells [26,27]. It is based on the refractive index (RI) of transparent samples, being particularly effective in monitoring adhered cells and microtissues such as spheroids [59]. This method provides quantitative phase images that map the optical phase delay induced by the sample in transmitted light, with pixel-based variations influenced by the sample’s RI and thickness [59]. This allows for the detailed characterization of cells, including morphological parameters such as area, optical volume, and optical thickness [60].

As a label-free method, holographic microscopy eliminates the need for staining, thus preventing adverse effects of cytotoxicity and phototoxicity, which are often associated with labeling techniques [29]. This aspect ensures the preservation of cell integrity, allowing the use of reduced light levels and safer conditions for the cells under study [61]. Despite these advantages, holographic microscopy presents some limitations that impact its resolving capabilities. These include the finite size of the pixels, which affects image quality, the need for a high sampling rate to avoid image reconstruction overlapping and the presence of speckle noise, which needs to be reduced for clearer imaging [62].

Nonetheless, holographic microscopy has proven relevant in tumor cell analysis, providing valuable insights into dynamic processes such as migration, proliferation, and apoptosis [63]. For example, Dubois et al. [64] used holographic microscopy to analyze the migration of HT-1080 fibrosarcoma cells in a 3D collagen gel, observing significant variation in migratory behavior, with distances ranging from 73 to 359 μm and speeds between 14 and 52 μm per hour. This variability indicated a high heterogeneity in the tumor cell population [64]. Additionally, holographic microscopy was employed to quantify tumor cell invasion by tracking parameters such as phase change, optical volume, and motility (Figure 5B) [65]. Notably, invasive HeLa cells exhibited a marked reduction in area and volume, correlating with cellular infiltration into the monolayer, while non-invasive cells retained their original morphology [65]. The study further revealed that cells with greater migratory capacity exhibited slower invasion speeds, underscoring the importance of cell movement dynamics in the tumor infiltration process [65]. Holographic microscopy has also effectively differentiated cells undergoing apoptosis from healthy control cells by offering precise quantitative measurements of cell number, volume, and confluency. It allows accurate distinctions of primary tumor cells from metastatic ones, giving insights into cancer progression and treatment responses [66,67].

For instance, studies integrating holographic microscopy with fluorescence correlation spectroscopy (FCS) enabled the detailed analysis of nucleolar RI and volume, shedding light on nucleolar dynamics under physiological stresses such as ATP depletion and transcriptional inhibition [36,37]. They demonstrated that the nucleolus has a dynamic rather than rigid structure, with significant variations in its morphology and volume under different physiological conditions [68,69]. Using holographic microscopy, 3D images showcased nucleolus heterogeneity, with different RI bands representing variations in molecular density. Specifically, the nucleolus showed an average RI of 1.363, higher than the RI of cytoplasm and nucleoplasm [68].

Bimodal analysis in holographic microscopy has further revealed optical biomarkers that differentiate cells with varying invasion and proliferation potential, such as between the B16F1 and B16F10 murine melanoma cell lines [70]. Specifically, cells with greater metastatic potential (F10) exhibited a higher refractive index (RI: 1.3989 ± 0.0112 µg/m^2^) and lower dry mass density (0.8003 ± 0.1771 µg/m^2^), while cells with lower metastatic potential (F1) showed a lower RI (1.3610 ± 0.0039 µg/m^2^) and higher dry mass density (1.4261 ± 0.0844 µg/m^2^). Furthermore, the phase distribution also changed. Cells with lower metastatic potential showed a multimodal distribution, with multiple peaks on the graph, indicating heterogeneity and distinct subpopulations. In contrast, cells with higher metastatic potential displayed a unimodal distribution, with a single peak, suggesting more uniformity and shared characteristics [70]. This is important for identifying cancer cell subpopulations with distinct invasive behaviors.

The technique has also shown great promise in enhancing the accuracy of urothelial carcinoma diagnosis in urinary cytology samples. Pham et al. used it to distinguish benign and malignant urothelial cells, with nuclear dry mass and nuclear entropy serving as differentiators [69]. Nuclear dry mass and nuclear entropy were significantly increased in cancer patients’ cells compared to healthy ones, showing an area under the curve (AUC) of 0.98 and 1, respectively, which indicated excellent classification accuracy [69]. The nuclear dry mass quantified the integrated mass of the entire nucleus, and an increased nuclear dry mass could be associated with higher DNA content [69]. The nuclear dry mass of healthy epithelial cells ranged from 15 to 17 picograms (pg), with an average of 15.8 pg. In contrast, cancer cells ranged from 33 to 35 pg, with an average of 34.4 pg. Notably, both nuclear dry mass and entropy increased progressively with malignancy. The average total cell masses were 51.1 pg for negative, 49.2 pg for atypical, 59.5 pg for suspicious, and 52.6 pg for positive groups. Nuclear entropy ranged from 0.35 to 0.80 in healthy cases, and from 1.00 to 1.55 in cancerous samples [69]. Similar applications have extended to live adherent pancreatic tumor cells, where holographic microscopy facilitated the measurement of cell morphology, RI, and cell response to drug treatments [71]. These cells had an average RI value of 1.38 ± 0.015, and thickness ranging from 7 to 23 ± 1 μm. As for treatment responses, after adding Latrunculin B toxin, cells had a decrease over 50% in thickness, highlighting the contribution of this technique to associate biological processes with biophysical cellular alterations [71].

In recent studies, complementary analytical techniques have increased the diagnostic power of holographic microscopy. For example, combining holographic microscopy with self-supervised learning enabled an accurate distinction of different cancer cell lines, by analyzing tumor cells’ morphology and intracellular content [72,73]. It efficiently identified different types of cell death, such as apoptosis and necroptosis, in cancer cells [66], opening new avenues for investigating regulatory mechanisms associated with cell death [66,72,73]. with the addition of Raman spectroscopy further enhanced, specifically between parental cells, CTCs, and lung metastatic cells [74]. The obtained 3D RI tomograms revealed morphological variations, such as area, aspect ratio, and dry mass, which increased with metastatic potential. Lung metastatic cells showed significantly higher dry mass compared to parental cells and CTCs. Yet, statistical analysis showed that while there were significant morphological differences, overlapping distributions made it difficult to classify cells based solely on these features. In contrast, Raman spectroscopy identified distinct spectral signatures for each cell type [74].

Beyond oncology, holographic microscopy has made a significant contribution in hematological monitoring. Here, optical diffraction tomography (ODT), a 3D extension of holographic microscopy—has been employed to classify and analyze blood cells in a label-free manner. (Figure 5C). A study by Ryu et al. integrated OTD with deep learning algorithms to classify blood cells, achieving high accuracy (0.971) in detecting normal and abnormal cells, plus estimating hematological indices [75]. These include mean corpuscular volume (MCV) and mean corpuscular hemoglobin (MCH), identified by analyzing the RI distributions of individual cells. These findings suggest that ODT, particularly when combined with artificial intelligence, offers a promising label-free alternative to traditional hematological analyzers [75].

In summary, holographic microscopy is an innovative, non-invasive technique for 3D analysis of living cells, which is advantageous for studying tumor dynamic processes and cellular morphology Its ability to quantify optical phase and cell morphology in real-time, along with its compatibility with other techniques, makes it indispensable in cancer research and beyond. However, challenges such as pixel size limitations, high sampling size, and the presence of noise must be addressed to improve its resolution and applicability.
Figure 5**Holographic microscopy-based imaging and analysis**. (**A**) Hologram formation and analysis pipeline. (i) Coherent incident light (light blue) interacts with scattered light from the specimen (dark blue), creating an interference pattern, known as the hologram. This pattern encodes information about the object in the central intensity profile. (ii) Representative examples of hologram-fitting models, and 3D reconstructions for a single microsphere, a sphere doublet, and a capsule-shaped bacterium. (iii) Hologram analysis enables quantitative extraction of biophysical parameters. Sequential acquisition over time allows tracking object motion in 3D or monitoring changes in physical properties. Reproduced with permission from [76]; (**B**) Comparative analysis of invasive and non-invasive HeLa cells. (i–v) 3D holographic cross-sections (50 µm) of individual cells at successive time points: (i) 20 min, (ii) 190 min, (iii) 240 min, (iv) 255 min, and (v) 270 min. Black arrowheads indicate invasive cells; red arrowheads with yellow outlines indicate non-invasive cells. A progressive decline in peak values and an increase in baseline levels suggest cellular infiltration. (vi) Quantification of area changes over time, with a decline reflecting invasion progression. Reproduced with permission from [65]. (**C**) Hematological analysis using ODT and a deep learning framework for automated single-cell analysis and classification. (i) Workflow overview of ODT analysis comprising (a) 3D RI tomogram of blood sample; (b) deep learning model and classification into four types: RBC, ARBC, PLT, and WBC; and (c) quantitative analysis based on RI values. (ii) 2D t-SNE plot illustrating the embedding of individual cells based on morphological features extracted from RI data. Cells are color-coded based on model predictions and true labels (RBC vs. ARBC), showing a continuous morphological transition. Representative RI tomographic slices of individual cells along this transition are shown below the plot. (iii) Box plots demonstrate the bias in Mean Corpuscular Volume (MCV) and Mean Corpuscular Hemoglobin (MCH) between conventional Complete Blood Count (CBC) analyzers and the proposed RI-based Quantitative Blood Analyzer (QBA) for RBCs. The RI-based method shows close agreement with standard clinical measurements while providing single-cell resolution. Reproduced with permission from [75].
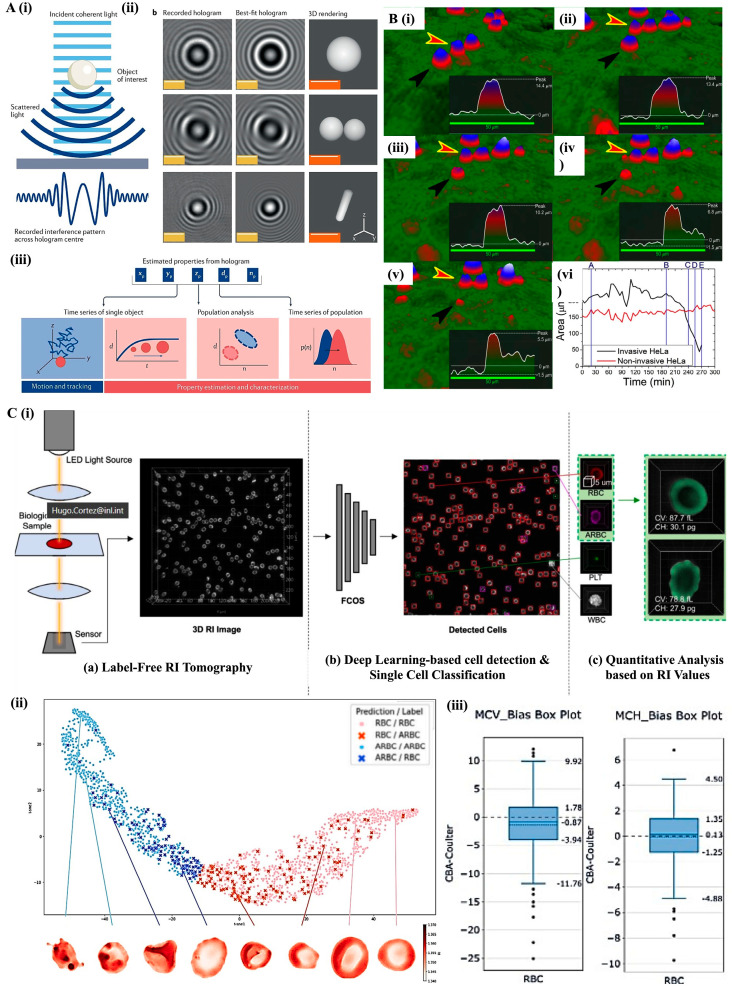



## 4. Cytometric Techniques

Cytometry encompasses a range of analytical methods designed to quantify the physical and chemical characteristics of individual cells or particles. Traditionally, these methods have relied on light scattering and fluorescence emission to evaluate parameters such as size, granularity, and biomarker expression [77]. Modern cytometric techniques have introduced label-free strategies that expand the scope of cellular features analyzed, such as electrical impedance, mechanical deformability, and morphology. In label-free cytometry, cells typically traverse laser beams or electrical fields, with detectors capturing real-time data on their intrinsic properties [78]. The following subsections explore several of these modalities that have shown promise in cancer research and diagnostics.

### 4.1. Deformability Cytometry: Mechanical Fingerprinting of Cells

Deformability cytometry allows the analysis of cells’ mechanical properties, such as deformability and viscoelasticity—features closely linked to key biological processes, such as cell differentiation, therapeutic response, and disease progression [79]. Real-time assessment of these biophysical properties provides valuable insights into dynamic cell behaviors under physiological and pathological states. Early deformability cytometry platforms used extensional flow to stretch cells, achieving throughputs of up to 1000 cells per second. Deformability was quantified by the maximum aspect ratio during forced deformation, offering greater throughput and sensitivity than previous techniques for studying cell deformation under stress [79]. Notably, one of the first applications of this approach involved using Latrunculin B to treat HL60 leukemia cells, revealing increased deformability after acting disruption. These mechanophenotypic changes distinguish treated and untreated cancer cells, highlighting the sensitivity of deformability cytometry to subtle cytoskeletal alterations, which other cytometric techniques might not detect [80]. A more recent development, viscoelastic deformability cytometry (vDC), was introduced by Asghari et al. [81] and improved upon earlier platforms by offering a high-throughput, label-free platform capable of analyzing up to 100,000 cells per second. This is achieved through parallelized microfluidics and viscoelastic flow (Figure 6A) to perform real-time mechanical phenotyping of both liquid and solid biopsies. By assessing cell deformability and size, vDC can distinguish malignant from normal cells. For example, RBCs and PBMCs differed in deformability, while breast cancer cell lines were distinguished by size. Importantly, vDC demonstrated its diagnostic potential by identifying chronic lymphocytic leukemia (CLL) cells, which exhibited significantly higher deformability than healthy B cells or PBMCs. This approach was also able to differentiate glioma-initiating cells (GICs) from more differentiated glioma cells and astrocytes, and it successfully detected rare cancer cells in blood, making it particularly useful for liquid biopsy applications [81]. These results established vDC as a robust, label-free method for cancer detection and disease monitoring via mechanical phenotyping.
Figure 6**Overview of viscoelastic and constriction-based deformability cytometry systems**. (**A**) Schematics of viscoelastic deformability cytometry ((i), vDC) device with three functional regions: R1 removes cell aggregates via a micropillar array; R2 aligns cells in parallel microchannels (3 cm, 50 μm × 50 μm) using viscoelastic focusing; R3 deforms cells in narrow constrictions (15 μm × 15 μm × 300 μm). CFD simulation shows wall shear stress (WSS) distribution, and time-lapse images capture cell deformation in R3. (ii) Density plots illustrate cell deformability versus size, aiding in subtype classification. Reproduced with permission from [81]. (**B**) Constriction-based deformability cytometry (cDC) system integrated with the ATMQcD deep learning framework. (i) Design of microconstriction arrays (10 μm × 30 μm × 60 μm) to ensure uniform velocity; (ii) high-speed imaging and analysis setup; (iii) time-resolved tracking of single-cell deformation. (iv) A quantitative comparison of MCF7 and MDA-MB-231 cells reveals differences in passage time, deformation index, and area-in-constriction, enabling accurate cancer cell classification. Reproduced with permission from [82].
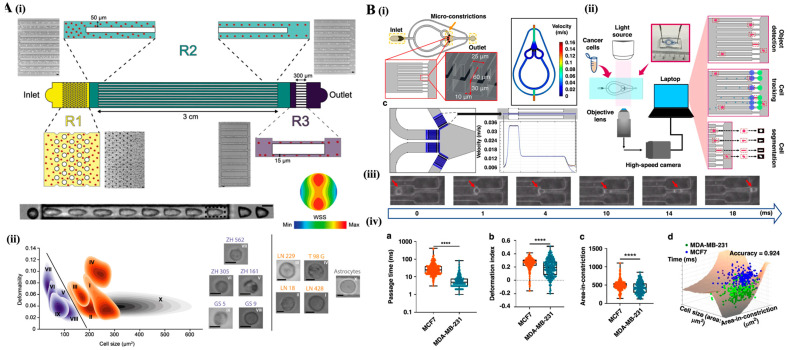



Further advancements in deformability cytometry include the development of constriction-based deformability cytometry (cDC) by Hua et al. [82]. Integrated with a deep learning–powered computational framework (ATMQcD), this new platform had a throughput of approximately 25,000 cells per minute. Although the rate is slower than earlier platforms (Figure 6B). cDC captures a broader range of biomechanical parameters, including passage time, area-in-constriction, and deformation index. These features are processed through a pipeline combining object detection (YOLOv5), tracking (Deep SORT), and segmentation (ResUNet++) and then used to extract stiffness index (c1), enabling quantification of cell deformability at single-cell resolution, independent of size. When applied to breast, lung, and bladder cancer cell lines, including hypoxia-conditioned subpopulations, cDC successfully stratified metastatic potential and identified mechanical heterogeneity within mixed populations [82]. Notably, it classified metastatic phenotypes with over 92% accuracy using a support vector machine model trained on multiparametric data and achieved 89.5% accuracy in distinguishing cancer cells from leukocytes in a blood-like context. The platform also incorporated a physics-based viscoelastic model grounded in power-law rheology, enabling stiffness estimation that aligned with atomic force microscopy benchmarks. In contrast, earlier systems were unable to achieve this level of stratification.

Collectively, the updated advances underscore the growing clinical and translational relevance of deformability cytometry. As a high-throughput, label-free modality for analyzing cellular mechanics, deformability cytometry is increasingly recognized for stratification cancer phenotypes, monitoring disease progression, and evaluating therapeutic response in both research and diagnostic settings.

### 4.2. Impedance Cytometry: Electrical Cell Characterization

Impedance cytometry provides a label-free method for detecting several cellular biophysical properties, such as size, morphology, viability and dielectric properties of individual cells [83]. This cytometry stands out for its ability to characterize cells without the complexities of optical methods, eliminating the need for high-transparency devices or frequent calibration of laser beams [83].

One of the key advantages of this method is its ability to distinguish between live and dead cells. For example, a microfluidic impedance cytometry (MIC) device, incorporating a novel double-differential electrode configuration, demonstrated a 94.5% accuracy in classifying live versus dead MCF-7 breast cancer cells, although only at a rate of approximately 1000 cells per second [84]. The system used low-cost 100 µm-wide indium tin oxide (ITO) electrodes and provided high-resolution measurements of electrical diameter and opacity metrics, non-invasive indicators of cellular viability (Figure 7A) [84]. Remarkably, the device also distinguished particles of different sizes, such as 7 µm and 10 µm polystyrene beads, achieving a geometry ratio between target and electrode width of 1:20, the lowest reported in impedance cytometry to date. This sensitivity and resolution, especially for small bioparticles and subtle dielectric shifts in cell viability shifts, position this technique at the center of sensitive and scalable ways of detection. Its affordability (under USD 1 per chip) and ease of fabrication with ITO electrodes instead of lithographically gold electrodes make it an even more attractive tool for point-of-care diagnostics, drug screening, and cell state monitoring in biomedical applications.

Recently, impedance cytometry has broadened its applicability from cancer cells to cancer-associated fibroblasts (CAFs), providing insight into these cell–cell interactions within the TME. In a recent study, multifrequency single-cell impedance cytometry (0.5–30 MHz) was used to differentiate cancer cells from cancer-associated fibroblasts (CAFs) in pancreatic ductal adenocarcinoma (PDAC) samples, with over 93% accuracy. This classification was based on impedance-derived metrics such as membrane capacitance, cytoplasmic conductivity, and electrical diameter (Figure 7B) [85]. Notably, this label-free method revealed that PDAC cells co-cultured with CAFs exhibited increased viability (around 85%) after gemcitabine treatment, compared to 50% viability in monoculture. This suggests that CAFs contribute to the reprogramming of cancer cells toward a drug-resistant phenotype. The system also demonstrated the ability to detect subtle shifts in cell populations in mixed samples and predict cell-type composition with high accuracy (up to 95%) after treatment, even when traditional markers like EpCAM were unreliable due to non-specific uptake by CAFs [53]. These findings underscore the potential of impedance cytometry, paired with machine learning, to analyze phenotypic transitions linked to chemoresistance and track tumor heterogeneity and cell state in a non-invasive, scalable manner, These findings underscore the potential of impedance cytometry, paired with machine learning, to dissect phenotypic transitions associated with chemoresistance and to track tumor heterogeneity and cell state in a non-invasive, scalable manner.

In an exciting recent development, a hybrid method combining microfluidic impedance cytometry with generative artificial intelligence (AI) was introduced by Kokabi et al. [86]. This involved capturing signals as cells passed through polydimethylsiloxane (PDMS) microchannels with integrated electrodes, followed by analysis through a deep neural network. The system successfully classified two cancer cell lines (MDA-MB-231 breast cancer and HeLa cervical cancer cells), with a 91% accuracy rate. More impressively, the AI-based model reconstructed high-resolution images of cancer cells from their electrical impedance profiles, achieving structural fidelity with mean structural similarity index (MSSIM) scores of 0.97 for breast cancer cells and 0.93 for control beads [86]. This impedance-AI hybrid approach represents a paradigm shift in label-free cellular analysis, offering a scalable and cost-effective alternative to traditional microscopy. By bridging biophysical profiling with qualitative imaging, this platform allows morphological assessment and classification from a single measurement modality, marking a significant advancement in cellular characterization.

However, a potential disadvantage of this approach is that the accuracy and structural fidelity of the AI-based model could heavily rely on training data quality. If the model is not trained with a sufficiently diverse set of cell types or conditions, it may struggle to generalize or misclassify cells outside the trained dataset. Moreover, the system might face challenges with highly heterogeneous or rare cell populations, such as CAFs or CTCs, where impedance signals alone may not provide enough distinct features for reliable classification [83].

Despite these limitations, continuous innovations position impedance cytometry as transformative, scalable, non-invasive platform with transformative potential for applications in drug screening, tumor microenvironment profiling, and real-time monitoring of phenotypic transitions in cancer and other diseases.
Figure 7Label-free impedance cytometry platforms for assessing cell viability and classifying tumor cell populations. (**A**) Low-cost impedance cytometry for label-free analysis. (i) Schematic of the double differential impedance cytometry setup using ITO electrodes and PDMS microchannels. The system captures multifrequency impedance signatures (magnitude and phase) as cells flow through the detection zone. (ii) Representative analysis of MCF-7 breast cancer cells showing distinct distributions in electric diameter and opacity between live and dead cells, enabling label-free viability discrimination with high throughput (~1000 cells/s). Reproduced with permission from [84] (**B**) Multifrequency impedance cytometry with supervised learning to classify pancreatic cancer cells and CAFs. (i) Workflow: PDAC tumor cells and CAFs derived from patient xenografts are co-cultured and subjected to gemcitabine treatment. Biophysical reprogramming of cancer cells toward drug resistance is assessed. (ii) Impedance phase data at multiple frequencies gates viable versus non-viable cells and allows comparison between monoculture versus. co-culture. (iii) SVM model classifies cancer and CAF populations based on multifrequency impedance data. Scatter plots show predicted class distributions across conditions with increasing drug resistance. Cancer cell prevalence decreases with treatment, while CAFs become dominant in mixed populations. Reproduced with permission of [85].
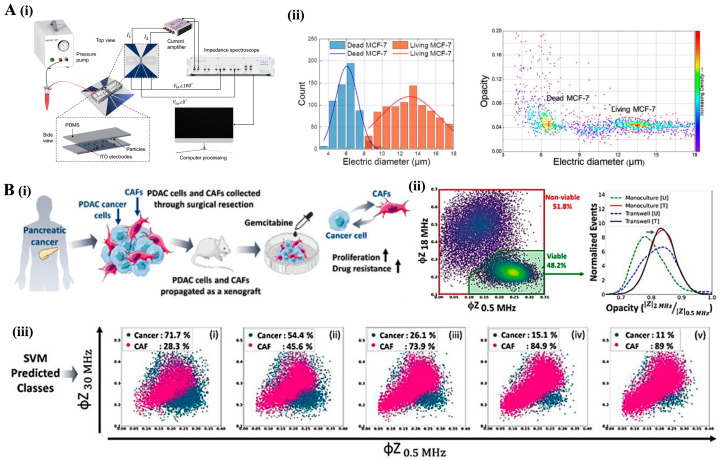



### 4.3. Imaging Flow Cytometry: High-Throughput Morphological and Functional Analysis

Imaging flow cytometry (IFC) combines the capabilities of traditional flow cytometry with the detailed morphological analysis of microscopy. By capturing high-resolution images at high-throughput, IFC enables single-cell analysis with precise morphometric cellular analysis, making it a powerful tool for label-free cancer cell detection [87]. Unlike conventional flow cytometry, which relies on fluorescent probes, label-free IFC eliminates issues like non-specific probe binding, interference, and low quantitative performance. It overcomes the inherent challenges of labeling, such as variability in probe binding and cellular molecule labeling, which can compromise detection accuracy [88]. Despite its advantages, label-free IFC is not without challenges. These include high data complexity, the need for advanced instrumentation, and difficulties with data storage and analysis. Additionally, the high-throughput nature of IFC can complicate equipment handling, and software limitations, such as manual region definition, still exist [89]. To address them, recent innovations in the field have focused on integrating complementary imaging modalities and AI-driven analysis to streamline data interpretation and enhance the precision of cell classification

### Multimodal Integration and AI-Enhanced Applications in IFC

Recent advancements in IFC have combined dual-modality imaging systems to capture a broader range of singular cellular features with greater efficiency. One example is the integration of brightfield and side-scattering images using a single detector, which allows faster, more accurate identification of cell subtypes based solely on light-scattering features. This system achieved an impressive 89.5% accuracy in classifying immune cell subtypes, demonstrating the potential of label-free approaches in immune profiling and disease monitoring [90]. Another significant breakthrough is the development of multiplexed asymmetric-detection time-stretch optical microscopy (multi-ATOM). This platform, which combines IFC with ultrafast quantitative phase imaging, can capture up to 10,000 cells per second, dramatically increasing throughput while maintaining subcellular resolution [91]. Multi-ATOM has been used to distinguish leukemia subtypes with an accuracy of 92–97%, highlighting its potential for hematological diagnostics and cell-based research [91].

Another promising application of IFC is its use in detecting bladder cancer cells in urine. A quantitative interferometric IFC (QIFC) system developed by Dudaie et al. [88] employed interferometric phase microscopy (IPM) to capture optical path delay (OPD) maps of individual cells as they flow through the system (Figure 8A). By segmenting cells using deep learning and machine learning algorithms, the system extracted 20 biophysical features, including morphology and OPD-related parameters such as dry mass and optical volume, to classify cells as benign or malignant. This platform achieved 96% accuracy and 96% AUC in clinical urine samples, outperforming some FDA-approved urinary biomarkers [88]. It demonstrated significant differences in cell shape, internal complexity, and dry mass between healthy urothelial cells and bladder cancer cells. Malignant cells had higher circularity and mean OPD, but lower area, OPD energy, OPD volume, dry mass, OPD entropy, and contrast compared to healthy cells (all *p* < 0.001). Similarly, deep learning-enhanced image cytometry (DLIC) has shown great promise in detecting altered PBMCs from extra nodal NK/T-cell lymphoma (ENKTL) patients [92]. By extracting 3 biophysical markers—cell size, eccentricity, and refractive index (RI), from over 270,000 PBMCs, DLIC revealed stage-specific phenotypic heterogeneity that correlated with relapse and treatment status. (Figure 8B). Mean size increased (80.9 μm^2^ to 81.9 μm^2^), eccentricity decreased (0.59 to 0.56), and RI slightly decreased (1.3915 to 1.3911) from the interim to end-of-treatment groups [92]. Their system achieved on-site, label-free quantification, laying the foundation for scalable, precision monitoring in hematologic malignancies for clinical decision support.
Figure 8Multimodal label-free optical cytometry platforms for biophysical profiling and classification of single cells in cancer diagnostics. (**A**) Quantitative interferometric IFC for bladder cancer detection. (i) Workflow for single-cell phase image acquisition using interferometric phase microscopy (IPM), followed by deep-learning-based segmentation and feature extraction. Optical path delay (OPD)-based features and morphological descriptors are used for classification. (ii) Representative OPD heatmaps of individual cells from healthy bladder epithelial cells (BdEC), red and white blood cells (RBC, WBC), and multiple bladder cancer cell lines, illustrating differences in biophysical properties like mass and optical thickness. Reproduced with permission of [88]. (**B**) Deep learning-integrated image cytometry (DLIC) for profiling PBMCs in extranodal NK/T cell lymphoma (ENKTL). (i) Schematic of treatment monitoring: PBMCs are profiled before and after treatment to extract biophysical traits (size, eccentricity, and refractive index, RI). Post-treatment cells exhibit reduced size and heterogeneity, with trends reflecting disease progression. (ii) 3D scatter plots show single-cell distributions of size, eccentricity, and RI in interim (IOT) and end-of-treatment (EOT) stages, revealing reduced cellular heterogeneity and a shift toward healthy-like profiles after treatment. Reproduced with permission of [92].
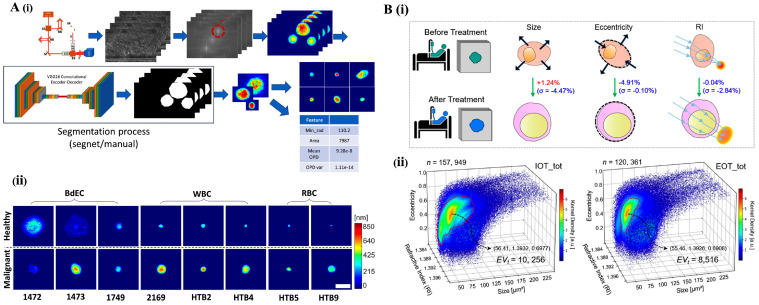



Another example is the integration of machine learning with IFC to predict DNA content and classification of cell cycle phases [93]. In fixed Jurkat cells, a high correlation was observed between predicted and actual DNA content (Pearson’s r = 0.896 ± 0.007), allowing classification into G1, S, and G2/M phases without fluorescence staining. Mitotic phase classification reached true positive rates of100% for anaphase and telophase, but 65.5% for prophase. This lower performance in prophase was attributed to its brief duration and subtle chromosomal changes, making it harder to distinguish from metaphase and increasing the chances of misclassification [93]. Nevertheless, the platform also effectively detected cell cycle perturbations, such as a 19% increase in G2/M-phase cells following a nocodazole treatment, confirming its utility in assessing drug-induced cell cycle arrest.

With these recent advancements, the integration of IFC with AI and other microscopy techniques has made IFC a promising technology for label-free cancer diagnosis. It stands out as a high-throughput cellular analysis with high-resolution imaging, surpassing limitations of conventional flow cytometry. However, integrating AI with IFC presents challenges, such as accurately interpreting complex data, handling short time point phases, and ensuring consistent performance. Addressing these challenges will be key to fully unlocking the potential of IFC for oncology.

## 5. Cell and Particle Scattering Techniques

Scattering-based biophysical techniques have emerged as powerful, label-free approaches for probing cancer cell properties and their surrounding microenvironment. By analyzing the interaction of light with cells, extracellular vesicles, and biomolecules, these methods provide valuable information on particle size, surface charge, and molecular composition, all of which are closely linked to tumor progression and metastatic potential. Unlike conventional biochemical assays, scattering techniques enable real-time and non-destructive measurements, often with minimal sample preparation, thus making them highly attractive for translational cancer research and diagnostics. The next section will address electro-mechanical and surface characterization techniques, which further extend our ability to probe cancer cell structure and function at the nanoscale.

### 5.1. Dynamic Light Scattering and Zeta Potential

Dynamic light scattering (DLS) is a label-free technique originally designed for characterizing particle sizes in suspension, particularly nanoscale particles and polymers. It relies on measuring temporal fluctuations in the intensity of scattered light, providing information about particle size distributions in a sample [94]. In contrast, static light scattering (SLS) measures the angular intensity distribution of the scattered light for larger particles or aggregates. Recently, DLS has been promising in cell and tissue analysis for cancer detection [95]. Notably, DLS is well-suited for exosomes analysis, nanoscale extracellular vesicles implicated in cancer biology due to their role in cell-to-cell communication. Strikingly, the zeta potential of exosomes secreted by cancer cells was demonstrated to be significantly more negative than that of non-malignant cells [96]. However, DLS presents some limitations. It is sensitive to environmental factors, such as temperature fluctuations and solvent viscosity, which can influence the data quality. Other challenges include low particle concentrations, signal-to-noise ratio issues, and artifacts like bubbles or reflective particles that reduce accuracy. These factors would need careful experimental control, larger sample volumes, and samples with higher RI to ensure reliable results [97]. In cancer research, DLS provides insights into cellular dynamics, such as aggregation tendencies and changes in cellular motility that correlate with cancer progression. Studies have shown how the zeta potential (surface charge) of cancer cells measured varies (Figure 9A), with the pH and ionic strength of the surrounding medium. For instance, studies the zeta potential of breast cancer cells becomes more negative at alkaline pH and neutralizes or reverses at acidic pH (Figure 9B) [98]. Thus, increased ionic strength, especially with divalent and trivalent electrolytes, reduces surface charge and promotes cellular aggregation, directly impacting tumor cell behavior and potentially metastasis [98]. Moreover, DLS has been used to detect triple-negative breast cancer (TNBC, Figure 9C), by evaluating intracellular motility and mitochondrial dynamics [95]. Correlating these findings with confocal fluorescent imaging, researchers classified TNBCs with high diagnostic accuracy (area under the curve of 0.95, accuracy of 0.89), highlighting DLS’s potential for precise, label-free cancer diagnostics.

Furthermore, DLS can be integrated with glycocalyx analysis, which governs cellular adhesion. Variations in adhesion, modulated by changes in glycocalyx, directly impact cancer cell metastasis. The removal of specific glycocalyx components, such as chondroitin sulfate, influences adhesion dynamics; intense removal reduces adhesion strength and speed, whereas moderate digestion enhances these parameters [99]. Real-time measurement of cell adhesion kinetics using Resonant Waveguide Grating (RWG) technology provides high-resolution insights into the interactions between cancer cells and the extracellular matrix [99]. Combining these approaches enables a deeper understanding of cancer cell behavior and metastatic potential.

Although zeta potential analysis in suspension cells is still not widely used, its relevance has been demonstrated above. This is currently more common in the study of extracellular vesicles, due to the nanometric size of these vesicles [96].

Overall, DLS has been showing promising applications as a non-invasive technique for cancer diagnosis, with relevance in cancer cells and extracellular vesicles analysis. Nonetheless, its current use in routine clinical practice remains limited, primarily due to challenges with low signal sensitivity and the need for high-quality samples. The integration with complementary techniques could make it a powerful tool for identifying biomarkers associated with cancer, particularly through the study of extracellular vesicles. Its capabilities make it a useful technique for investigating cellular interactions, with significant potential for identifying biomarkers associated with cancer.
Figure 9**Dynamic light scattering (DLS) and zeta potential analyses**. (**A**) Zeta potential (ζ-potential) values of different mammalian cells and bacteria. (i) Suspension cells generally exhibit more depolarized ζ-potentials than adherent ones. Cellular activation (e.g., platelets, macrophages) and transformation (e.g., MCF-10A to MCF-7) are associated with ζ-potential depolarization, while differentiation (e.g., iPSCs to neurons) shows the opposite trend. Among bacteria, ζ-potential varies by species and viability, with dead cells having distinct values due to altered surface charge. The variations underscore the potential of ζ-potential as a label-free indicator of cell state and identity. Reproduced with permission from [100]. (**B**) Zeta potential characterization. (i) Influence of zeta potential on two normal cell lines and breast tumor cells in a Na^+^ solution with an ionic strength of 10^−2^ M. Tumor cells exhibit the most negative zeta potential, reaching approximately −15 mV at neutral pH, while normal cells range between −20 mV and −25 mV at the same pH. (ii) Zeta potential of normal and tumor cells in the presence of trivalent ions (iron and aluminum) at pH 4 across three ionic strengths. Tumor cells display less negative zeta potential values than normal cells under all conditions, with iron suggesting lower absorption than aluminum in tumor cells. Reproduced with permission from [98]). (**C**) DLS-based intracellular dynamics. (i) Time-dependent fluctuations in scattered light intensity at the central point of confocal DLS images for MDA-MB-231 and SKBR3 breast cancer cells. (ii) Average autocorrelation functions derived from DLS images for both cell types. Experimental data points are shown as blue (MDA-MB-231) and red (SKBR3) dots. The slower decay in SKBR3 cells indicates reduced motion of intracellular scattering particles. Reproduced with permission from [95].
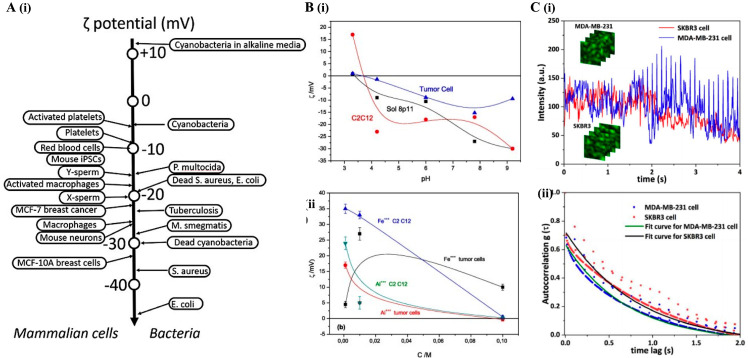



### 5.2. Surface-Enhanced Raman Spectroscopy

Surface-enhanced Raman spectroscopy (SERS) stands out in its high specificity and sensitivity for detecting biomarkers at very low concentrations and the identification of CTCs [101,102,103]. By exploiting the enhanced Raman scattering from molecules adsorbed onto metal surfaces, SERS offers high-resolution molecular signatures, enabling the differentiation of cancerous from non-cancerous cells. Particularly, SERS can identify disease-associated alterations in serum, plasma or urine samples [101]. Additionally, SERS can be applied for the characterization of CTCs, a rare and challenging target for conventional detection methods [102]. However, this technique has limitations, including the complexity involved in producing reliable plasmonic substrates and the need for more robust nanoparticles to ensure reproducibility. Nonetheless, ongoing advancements in nanoparticle design and substrate development are continuously improving the performance and reliability of SERS for clinical applications [103].

One of the major advantages of SERS lies in its ability to identify serum biomarkers with diagnostic accuracies exceeding 90% [101]. Specifically, research into the effects of varying silver nanoparticle (AgNP) concentrations demonstrated significant spectral changes directly related to kidney cancer-specific biomarkers, notably hypoxanthine and uric acid [101]. Moreover, post-operative serum profiles have been shown to return to a baseline that closely resembles those of healthy individuals, demonstrating SERS’s utility in monitoring the efficacy of treatments like tumor removal [101]. evaluating treatment efficacy. In addition, SERS excels in the detection of CTCs, bypassing the need for complex cell separation processes, which is particularly beneficial in liquid biopsies, a less invasive method compared to traditional tissue biopsies [102].

The integration of SERS with microfluidic devices further enabled real-time, non-invasive analysis of individual tumor cells, offering a platform for detailed molecular profiling and discovery of novel cancer biomarkers [103]. Recent innovations have also integrated SERS with AI systems for earlier cancer detection. A study demonstrated that using only 15 µL of serum with silver nanowires to enhance the SERS signal provided a major advantage by reducing sample requirements, making the method less invasive and more suitable for clinical use. The resulting spectra were analyzed with a support vector machine (SVM) model across 50 Raman intervals (600–1800 cm^−1^), achieving excellent diagnostic performance with an accuracy of 95.81%, sensitivity of 95.40%, and specificity of 95.87% [104]. The study included over 1900 samples across five cancer types, namely lung, colorectal, hepatic, gastric, and esophageal, enhancing model robustness. This method also distinguished early cancer from other diseases and showed potential in identifying subtle spectral differences between cancerous and non-cancerous samples. These results highlight the feasibility of applying SERS-AI systems as a tool capable of adapting to different cancer types or patient needs.

In conclusion; SERS has remarkable potential in cancer diagnostics due to its high sensitivity and specificity. Its application in liquid biopsies and integration with AI systems provides a robust biophysical profile of cancer cells. Despite ongoing challenges related to substrate robustness and particle consistency; ERS is poised to become an indispensable tool in clinical oncology; particularly when combined with microfluidic technologies for single-cell analysis. The next section will cover atomic force microscopy (AFM) and electrical impedance spectroscopy (EIS). AFM and EIS are key techniques that complement scattering methods by providing detailed insights into cell structure and conductivity; essential for understanding all cancer cell biophysical features.

## 6. Electro-Mechanical and Surface Characterization

In addition to scattering methods, techniques that probe the mechanical and electrical properties of cells provide complementary insights into cancer biology. These approaches capture how cancer alters cell stiffness, elasticity, adhesion, and membrane conductivity, features closely linked to tumor aggressiveness and therapy response. Atomic Force Microscopy (AFM) enables nanoscale assessment of cell mechanics and surface structure [105] while Electrical Impedance Spectroscopy (EIS) evaluates membrane capacitance and resistance to monitor changes in cell physiology [106]. Together, these methods expand our understanding of tumor biomechanics and electrochemical signatures, offering potential for early diagnosis, prognosis, and personalized treatment strategies.

### 6.1. Atomic Force Microscopy

AFM is a versatile technique that provides characterization, manipulation and exploration of cellular surfaces with molecular precision at the nanoscale, as well as the dynamics of membranes. AFM operates by scanning the cell surface with a sharp probe, enabling the measurement of both structural organization and mechanical properties, such as adhesion and stiffness, under near-physiological conditions [105]. This minimal sample preparation, typically limited to fixation to the surface, allows AFM to study cells without extensive modification, preserving the integrity of biological features. Moreover, AFM’s sensitivity to vertical displacements allows for precise measurements of cellular mechanics, such as the elasticity and deformability of individual cells [107]. As for limitations, it has low throughput, a restricted imaging area, and relatively low spatial resolution compared to other microscopy techniques. These challenges may limit its application in high-throughput screening or large-scale analyses [108].

Over the last years, AFM has been invaluable for characterizing the mechanical properties of cancer cells (Figure 10). Specifically, malignant breast cancer cells were found to have altered mechanical characteristics such as increased deformability and reduced stiffness compared to non-malignant cells [107,108]. Similarly, AFM measurements of Young’s modulus of ovarian cells reveal that non-malignant ovarian IOSE cells had the lowest stiffness (2.47 ± 2.05 kPa), followed by HEY cells (0.88 ± 0.53 kPa), and the highest stiffness in HEY A8 cancer cells (0.49 ± 0.22 kPa), indicating a correlation with metastatic potential. Raji lymphoma cancer cells had lower stiffness (0.2 to 0.4 kPa) compared to non-cancerous T-lymphocytes (1 to 1.4 kPa) and K-562 bone marrow cells (0.6 to 0.7 kPa), suggesting that more aggressive cells are more deformable [109,110].
Figure 10**Images of electro-mechanical and surface characterization techniques**. (**A**) (i) Scheme of the inverted optical microscope; (ii) Graphical analysis of the elasticity of living bladder cells about indentation depth. Results are expressed as mean ± standard deviation. The elasticity of malignant cells is significantly lower than that of non-malignant cells and, in addition, the indentation depth influences the elasticity measurement; and (iii) Graphical analysis of variation in cell elasticity before and after exposure to cytochalasin D (Cyt-D). Results are expressed as mean ± standard deviation. The treatment resulted in a decrease in Young’s modulus, indicating the predominant influence of actin filaments on the mechanical properties of cells (Reproduced, with permission, from [111]); (**B**) (i) High-resolution images showing the distinct contrast patterns observed for each cell line analyzed: MCF-10A (healthy), MCF-7 (cancerous/non-invasive), and MDA-MB-231 (cancerous/invasive). The first column (a,d) of images corresponds to healthy cells, while those in the second column (b,e) refer to non-invasive cancer cells, and those in the third column (c,f) to invasive cancer cells. The images in the first row (a–c) illustrate the topographic contrast, while those in the second row (d–f) highlight the contrast in Young’s modulus. In all cases, the field of view of the images is 25 × 25 μm. The amplitude of topographic variation is 2 μm for MCF-10A, 4 μm for MCF-7 and 6 μm for MDA-MB-231. Images representing Young’s modulus are displayed in logarithmic color scale, covering values between 2.5 and 250 kPa.; and (ii) Comparative analysis of average Young’s modulus values for individual cells obtained by different indentation methods. Low load rate indentations at 1 Hz were used, employing sharp pyramidal tips and large radius spherical tips, in addition to the peak force modulation technique at 250 Hz with sharp pyramidal tips. Each point on the graph corresponds to the average value of the apparent Young’s modulus of a specific cell (Reproduced, with permission, from [112], being B (ii) from supporting information article section).
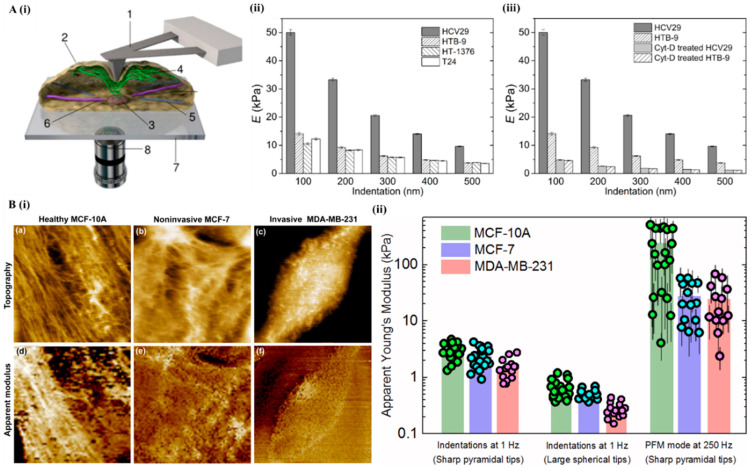



The deformability of cancer cells, assessed by AFM, is also significantly higher than that of non-malignant cells (Figure 10A), serving as a biomarker of bladder cancer cells [111]. Healthy bladder cells (HCV29) had a higher modulus of elasticity, ranging from 50 kPa at 100 nm to 10 kPa at 500 nm, compared to cancerous cells (HTB-9), which ranged from 15 kPa at 100 nm to 5 kPa at 500 nm, confirming that healthy cells are stiffer than cancer cells [111]. This was also verified in breast cancer cell lines, due to well-organized actin fibers (Figure 10B). At 1 Hz, cell stiffness decreased with malignancy: MCF-10A (0.7 ± 0.3 kPa), MCF-7 (0.5 ± 0.1 kPa), and MDA-MB-231 (0.3 ± 0.1 kPa). At 250 Hz, all cells became stiffer, but the stiffness difference between healthy and cancer cells increased: MCF-10A (250 ± 100 kPa), MCF-7 (28 ± 12 kPa), and MDA-MB-231 (25 ± 13 kPa) [112].

AFM is also valuable for probing tumor tissue mechanics, where it can distinguish between softer cancer cells and stiffer tumor stroma [113]. Tumors often have bimodal elasticity profiles- one peak representing soft cancer cells and another for the stiffer ECM. They also tend to be stiffer than healthy tissues due to increased ECM deposition, which can promote cancer cell migration and invasion. In contrast, normal and benign tissues showed a unimodal distribution, indicating less stiffness variability [113]. Furthermore, AFM has been used to study the heterogeneity of tumor spheroids, where variations in mechanical properties influence therapy response [114].

Overall, AFM’s ability to probe nanoscale mechanical properties provides essential insights into tumor biomechanics, offering potential for early cancer diagnosis and therapeutic assessment, though its application remains limited by technical constraints such as low throughput.

### 6.2. Electrical Impedance Spectroscopy

Electrical impedance spectroscopy (EIS) is a technique that has been applied for the early detection of neoplasms [106]. It measures the resistance and capacitance of cell membranes at varying frequencies, providing insight into the cell’s internal structure, size, and viability [106]. Unlike Zeta Potential measurements, which focus on surface charge, EIS assesses the overall impedance of the cell, offering a broader picture of its physiological state. However, this technique faces challenges in microfluidic devices due to the need for high sensitivity in detecting signals in small-volume sensors [115]. Additionally, the large volume of data generated, and potential interference require advanced statistical and computational methods for proper analysis [116].

Nonetheless, EIS has proven effective in spotting cancer by comparing the electrical properties of cancerous and healthy cells [106]. Comparing bioimpedance tissue measurements in breast cancer patients, the results showed lower impedance values for cancerous tissues compared to healthy ones. Its sensitivity was 77.8%, much higher than traditional breast cancer methods like mammography, which has a false negative rate of 4% to 34%. Cole-Cole parameters, which are used in EIS to represent the frequency-dependent dielectric properties (permittivity and conductivity). It was observed that in seven out of nine patients, cancerous tissues consistently exhibited lower internal resistance and membrane capacitance compared to healthy tissues [106].

At the single-cell level, EIS has also identified cancer cells by distinguishing between various stages of cancer progression [117]. EIS analysis of individual cells showed that normal breast MCF-10A cells had a membrane capacitance of 1.94 ± 0.14 AF/cm^2^ and resistance of 24.8 ± 1.05 MΩ at 100 kHz. Meanwhile, increasing metastatic cancer cell lines showed progressively lower capacitance: 1.86 ± 0.11 for MCF-7, 1.63 ± 0.17 for MDA-MB-231, and 1.57 ± 0.12 for MDA-MB-435. Compared to healthy cells, membrane capacitance decreased by 4.1%, 16.0%, and 19.1%, respectively. Electrical resistance remained similar, from 24.8 ± 0.93 MΩ for MCF-7 to 26.2 ± 1.07 MΩ for MDA-MB-435 [117]. Nevertheless, the decrease in capacitance suggests that as cancer progresses, the cell membrane becomes more permeable, which is an EIS measurement that could be used as an indicator of disease stage.

EIS has also shown value in identifying metastatic potential and monitoring chemoresistance, the ability of cancer cells to resist chemotherapy, by detecting resistant cells before symptoms manifest [118,119]. Less metastatic cells exhibited a significantly higher impedance phase value of −63.4° ± 8.6°, while highly metastatic cells showed a lower value of −73.4° ± 10.4° (*p* < 0.001) [118].

The technique also monitored drug-induced cell adhesion, proliferation, and death, revealing a decrease in impedance as cells became chemoresistant, indicating changes in their electrical properties. Additionally, EIS facilitated the identification of cellular plasticity. This early detection is crucial for personalized cancer treatment strategies, allowing for timely intervention [119].

In conclusion, EIS is a powerful tool for cancer detection by monitoring changes in electrical properties in cancer cells. Overcoming the challenges of sensitivity in small-volume sensors and improving accuracy will be crucial for expanding the clinical use of EIS. Then, features, such as capacitance and resistance, can indicate cancer progression, metastatic potential, and chemoresistance. EIS also allows for the monitoring of cellular plasticity, making it possible to identify drug-resistant cells before clinical symptoms appear. These capabilities highlight EIS’s potential for early diagnosis and personalized treatment strategies in cancer management.

## 7. Clinical Translation

The clinical translation of label-free biophysical cellular detection methods marks a significant step toward non-invasive, real-time, and personalized cancer diagnostics. These technologies, ranging from optical, mechanical, electrical, and microfluidic platforms, allow the assessment of intrinsic physical and functional characteristics of cells without the need for molecular labels, complex preparation steps, or cell destruction. Yet, several challenges must be addressed for widespread clinical adoption (Table 1).

One major limitation is the lack of standardization across platforms, including variation in chip design, measurement parameters, and cell processing protocols. This variability complicates comparison between studies and hinders reproducibility. Furthermore, most studies are limited to small patient cohorts or preclinical models. Therefore, there is a need for large-scale, multicenter validation studies to establish diagnostic accuracy, sensitivity, and clinical utility. Currently, many label-free platforms also lack cross-institutional reproducibility and standardized operating procedures, which hinders their integration into routine diagnostic workflows [120]. These systems must demonstrate consistent performance across diverse laboratory settings and patient populations for regulatory approval from agencies, namely the FDA or EMA.

Another significant challenge is sample throughput and automation. For label-free systems to be viable in busy clinical environments, they must be able to process large volumes of samples efficiently and reliably. For this, automation of image and data analysis, supported by AI and machine learning algorithms [121], will be crucial to minimize operator dependency, reduce errors, and improve diagnostic speed and accuracy.

Cost and accessibility are other critical factors for clinical translation. Although label-free technologies eliminate the recurring costs of biochemical labels and reagents, the underlying instrumentation- such as holographic microscopes, impedance cytometers, or imaging flow cytometers- can be expensive. Their complexity often demands skilled operation and regular maintenance. To overcome these limitations, future developments must focus on creating affordable, portable, and user-friendly systems. This is especially important for deployment in low-resource settings, where advanced diagnostic capabilities are often most needed but least available.

A key strength of label-free technologies lies in their potential for multiparametric integration. Techniques like imaging flow cytometry and surface-enhanced Raman spectroscopy (SERS) can be combined with AI and multimodal data analysis to provide comprehensive, label-free phenotyping of cellular and subcellular structures. This integration is invaluable in oncology, where it supports the characterization of CTCs, extracellular vesicles, and other biomarkers for personalized medicine.

Perhaps the most immediate and impactful clinical application of label-free technologies is in the field of liquid biopsy. These non-invasive diagnostic approaches use body fluids like blood, urine, or saliva, to detect cancer-related biomarkers. Label-free systems that isolate and analyze CTCs, cell-free DNA (cfDNA), or exosomes offer a powerful alternative to traditional tissue biopsies [122], by enabling early cancer detection, real-time monitoring of disease progression, and evaluation of therapeutic responses with minimal discomfort or risk to the patient. Additionally, by eliminating the need for chemical labeling, these methods reduce assay complexity and variability, making them suitable for repeated or longitudinal testing.

In summary (Figure 11), the clinical translation of label-free detection technologies is still a work in progress with the challenges of standardization, automation, cost reduction, and clinical validation to overcome. However, as these methods improve, especially through integration with AI and microfluidic systems, they hold great promise to revolutionize cancer diagnostics, offering precise, accessible, and non-invasive solutions for early detection and personalized treatment monitoring.
Figure 11Summary of the reviewed techniques, categorized by mechanical, electrical, light, and morphological features. Colors represent applicable cell form: yellow for adherent cells, blue for cells in suspension, and green to represent both. The differentiation of resolution is established by geometric shapes. Circles correspond to single cells and triangles indicate single cells and subcellular structures. Increased throughput is associated with greater repetition of these shapes, while reduced throughput is represented by a single circle or triangle. (Adapted, with permission, from [79]).
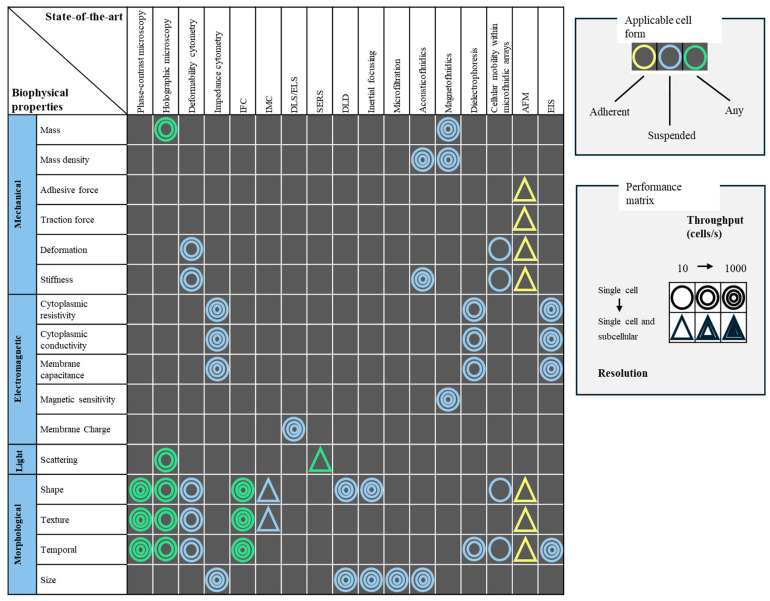



## 8. Conclusions and Future Perspectives

Label-free technologies that assess biophysical cellular properties have emerged as powerful tools in cancer research and diagnostics. They offer a unique perspective into cell behavior without the need for molecular labeling or extensive sample pre-processing. By leveraging intrinsic biophysical cues, such as cell size, shape, stiffness, dielectric properties, and surface adhesion, these techniques enable real-time, non-invasive characterization of cancer cells vs. normal cells, including rare populations such CTCs. As highlighted in this review, several platforms, ranging from optical imaging (e.g., holographic and phase-contrast microscopy), electro-mechanical profiling (e.g., impedance cytometry, atomic force microscopy), and microfluidic technologies (both passive and active), have demonstrated substantial potential in detecting, isolating, and phenotyping cancerous cells.

Despite their promising applications, clinical adoption of label-free biophysical methods faces several challenges. Among these are the lack of standardized protocols, limited multicenter validation studies, and technical issues related to reproducibility and device integration. Variability in measurement outputs across different platforms and sample types makes it difficult to establish consistent, universally accepted diagnostic thresholds. Furthermore, while AI-powered data analysis offers tremendous potential, its clinical deployment requires transparent, interpretable models supported by rigorous regulatory frameworks to ensure trust and accountability in medical settings.

Looking ahead, future research should prioritize the development of robust, hybrid platforms that combine multiple biophysical modalities to increase diagnostic specificity and sensitivity. Advancing towards compact, user-friendly and point-of-care systems will also be crucial to translating these technologies into routine clinical practice. In addition, more longitudinal studies with large patient cohorts are needed to validate the prognostic and predictive value of biophysical biomarkers over time. Nevertheless, collaborative efforts between engineers, biologists, clinicians, and data scientists are essential to translate technological innovations into clinical utility.

## Figures and Tables

**Table 1 bioengineering-12-01045-t001:** Overview of biophysical label-free techniques, their challenges and research gaps for clinical translation.

BIOPHYSICAL LABEL-FREE METHOD	TECHNIQUE	CHALLENGES	REQUIREMENTS FOR CLINICAL TRANSLATION
**MICROFLUIDIC SYSTEMS FOR CELL SEPARATION AND ENRICHMEMT**	Passive Cell Separation Microfluidics	DLD: Limited to size-dependent discrimination; Not sensitive to cellular shape or deformability. IF: Does not separate cells with similar biophysical features. MF: Shear stress damage; Not indicated for small volumes of samples	Miniaturization of DLD and IF chips. Integration with active sorting or functional assays. Improve resistance of chips to clogging. Enhance capture of cells within mixed populations and similar features. Improve detection for small volumes.
	Active Cell Separation Microfluidics	AC: Cell misalignment upon entry; potential for clogging; loss of rare cells post-separation processing MG: More expensive micromachining DC: Low single-cell resolution	Combine AC with DC Improve pre-alignment strategies. Modeling device geometry, tilt angles and change operational parameters Reduce fabrication costs of chips based on MG Fine-tune parameters for each cell type and sample matrix
	Cellular Mobility within Microfluidic Arrays	EICS: Maintaining high cell viability; Only for metastatic potential.	Study cell migration across cancer stages and types. Target other features, besides cell behavior;
**MICROSCOPY FOR CELLULAR ANALYSIS**	Phase-Contrast Microscopy	Limited to 2D imaging; Does not quantify subtle changes in depth; Segmentation errors in overlapping cells.	Create algorithms for automated validation. Refine segmentation in heterogeneous populations: Eliminate the need for time-consuming manual validation of results. Improved quantification of optical thickness.
	Holographic Microscopy	Requires high computational power for 3D reconstruction; sensitivity to noise	Address limitation in pixel size. Minimize the need of large sampling sizes. Enhance the ability of real-time monitoring of OPD and cell morphology.
**CYTOMETRY**	Deformability Cytometry	vDC: Only assess cell deformability depending on size. cDC: Lower throughput than vDC;	Increase throughput rate of cDC close to vDC rates. Turn vDC less dependent on particle size to stratify more cancer phenotypes.
	Impedance Cytometry	Sensitivity issues with small volumes and rare populations; Only relies on impedance feature	Increase the quality and diversity of training data for AI-based integration. Comparative studies of accuracy with traditional markers. Multimodal analysis with other features
	Imaging Flow Cytometry	High cost and complexity of systems; Requires expert handling of multimodal data; software limitations:	Integrate with AI to streamline data interpretation. Eliminate the need for manual region definition;
**CELL AND PARTICLE SCATTERING ANALYSIS**	Dynamic Light Scattering	Sensitive to temperature, pH …; Difficulty in differentiating between similar particles; requires high-quality and volume samples:	Combine it with glycocalyx analysis or confocal fluorescent imaging. Study the precise impact of any change on experimental conditions
	Surface-Enhanced Raman Spectroscopy	Requires high signal enhancement; Sample damage with intense laser use.; Low throughput.	Integrate with microfluidic technologies to increase throughput. Optimize formulation of plasmonic substrates and nanoparticles;
**ELECTRO-MECHANICAL AND SURFACE CHARACTERIZATION**	Atomic Force Microscopy	Slow data acquisition time; Difficulty in analyzing complex samples	Increase imaging area and spatial resolution;
	Electrical Impedance Spectroscopy	Sensitivity challenges in microfluidic devices with small sample volumes; Need of advanced statistical modules.	Increase sensitivity of impedance in small-volume sensors for scalable applications.

Abbreviations: DLD, Deterministic lateral displacement; IF, Inertial Focusing; MF, Microfiltration; AC, Acoustofluidics; MG, Magnetofluidics; DC: Dielectrophoresis; EICS, Electrical cell–substrate impedance sensing; vDC, Viscoelastic deformability cytometry; cDC, Constriction-based deformability cytometry.

## Data Availability

No new data were created or analyzed in this study. Data sharing does not apply to this article.

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
