# Peer review of "Label-Free Cancer Detection Methods Based on Biophysical Cell Phenotypes"

_bioengineering, 2025, doi:10.3390/bioengineering12101045_

Round 1

Reviewer 1 Report

Comments and Suggestions for Authors

This review manuscript provides a comprehensive and technically rich synthesis of label-free cancer detection methods based on biophysical cell phenotypes. The authors have successfully gathered an extensive range of optical, mechanical, electrical, and microfluidic approaches, offering readers a detailed view of the state-of-the-art and emerging technologies. The breadth of coverage, from microscopy techniques to advanced microfluidics and integrated AI-assisted systems, is a key strength, and the manuscript’s extensive referencing and illustrative examples support its credibility. 

That said, the manuscript’s scope and depth come at the cost of clarity and cohesion. The density of technical detail and the frequent transitions between very specific experimental findings make it challenging for readers to maintain a coherent narrative. In many sections, individual study results are described at great length without sufficient synthesis to distill overarching trends, limitations, and comparative advantages. This issue is most apparent in the middle sections, where highly specific performance metrics dominate the discussion but are not consistently tied back to the review’s stated objectives or to clinical applicability. Figures are generally informative, but their integration into the text could be improved by explicitly linking them to the broader arguments rather than treating them as stand-alone summaries.

There is also a need to sharpen the manuscript’s critical analysis. While the review is exhaustive in reporting technical advances, it is less rigorous in identifying knowledge gaps, unresolved technical challenges, and realistic barriers to clinical adoption. Many technologies are described in highly favorable terms without sufficient discussion of their reproducibility, scalability, and comparative cost-effectiveness, which are essential considerations for readers evaluating translational potential. The clinical translation section is a step in the right direction, but it would benefit from a more integrated critical appraisal across all techniques, not just at the end. Furthermore, some sections could be streamlined by reducing repetitive methodological descriptions and focusing instead on cross-comparisons that reveal where certain methods excel or fall short.

In terms of language and structure, the manuscript would benefit from reorganization to create a stronger thematic flow. Grouping techniques more explicitly by the type of biophysical property measured and integrating cross-cutting themes, such as throughput, invasiveness, and compatibility with AI, would help readers see connections across modalities. Attention to sentence concision and the removal of redundancies would improve readability without sacrificing detail.

Overall, this is a valuable and well-researched review that could become a highly cited reference in the field. However, it requires substantial structural and analytical refinement to reach its full potential. I recommend major revisions focused on synthesizing findings, deepening critical evaluation, and improving narrative coherence, while preserving the rich technical detail that is its hallmark.

Author Response

Ref. No.:  bioengineering-3809995

First of all, we thank the reviewers for their valuable feedback to improve the proposed review.

Based on the feedback from both reviewers, we reorganized the review to enhance clarity and improve the narrative flow. As Reviewer 2 suggested, “First, the microfluidics section seems to be somewhat out of place. Potentially, this section could become the first one since separation and enrichment would precede the characterizations.” We moved the microfluidics section to the beginning to align with this suggestion, ensuring that the review follows a more logical sequence. Here follows the new proposed index:

  1. Introduction
  2. Microfluidic Systems for Cell Separation and Enrichment (moved earlier, aligning with Reviewer 2’s suggestion)
    2.1. Passive Cell Separation Microfluidics
     2.1.1. Deterministic Lateral Displacement
     2.1.2. Inertial Focusing and Centrifugal Microfluidics
     2.1.3. Microfiltration
    2.2. Active Cell Separation Microfluidics
     2.2.1. Acoustofluidic Separation
     2.2.2. Magnetofluidics
     2.2.3. Dielectrophoresis Microfluidics
    2.3. Cellular Mobility within Microfluidic Arrays
  3. Label-Free Microscopy Techniques for Cellular Analysis
    3.1. Phase-Contrast Microscopy: Morphological Profiling and Segmentation
    3.2. Holographic Microscopy: Quantitative Phase Imaging for 3D Characterization
  4. Cytometric Techniques
    4.1. Deformability Cytometry: Mechanical Fingerprinting of Cells
    4.2. Impedance Cytometry: Electrical Cell Characterization
    4.3. Imaging Flow Cytometry: High-Throughput Morphological and Functional Analysis
     4.3.1. Multimodal Integration and AI-Enhanced Applications in IFC
  5. Cell and Particle Scattering Techniques
    5.1. Dynamic Light Scattering and Zeta Potential
    5.2. Surface-Enhanced Raman Spectroscopy
  6. Electro-Mechanical and Surface Characterization
    6.1. Atomic Force Microscopy
    6.2. Electrical Impedance Spectroscopy
  7. Clinical Translation
  8. Conclusions and Future Perspectives

Concerning Reviewer 2's suggestion and Reviewer 1's emphasis on structural improvements, we proceeded with significant revisions to enhance the clarity and flow of the review. Reviewer 1 pointed out that "the study results are described at great length without sufficient synthesis to distill overarching trends, limitations, and comparative advantages," and recommended "substantial structural and analytical refinement" to strengthen the review. To address these concerns, we worked on effectively synthesizing the described literature, focusing on a critical evaluation, and improving the narrative coherence. Nevertheless, we maintained the technical detail that characterizes the review. The following revisions were made across each section to improve clarity and the overall flow:

  1. Introduction (Pages 1-2)
    • The introduction was revised to improve clarity. Its structure was improved by previewing the techniques early, addressing Reviewer 2’s suggestion for better organization. Line 35 – 66 “Cancer, marked by abnormal and uncontrollable cell growth, encompasses a wide range of pathologies originating from almost any organ or tissue. [1] As the second leading cause of mortality globally[1], cancer´s poor prognosis and limited survival rates are often linked to late-stage diagnosis and advanced disease progression. [1,2] Conventional diagnostic methods, such as magnetic resonance imaging, biopsy histology, tomography, and invasive imaging (e.g., cystoscopy), [3] although effective, remain expensive and require sophisticated equipment and specialized clinicians. Moreover, these methods can be limited in resource-constrained settings and have inherent invasiveness. [4,5] Therefore, it is imperative to develop complementary and alternative non-invasive methodologies that are cost-efficient and easily automated for cancer diagnosis and prognosis. In this context, biophysical biomarkers such as extracellular matrix stiffness and cellular mechanical cues have gained interest as potential indicators for detecting cancer. These biomarkers influence key cellular processes such as differentiation, growth and motility of tumour cells. [8,9] For the identification and analysis of biophysical changes, advanced technologies, which will be mentioned later, are successfully used for macroscopic tumour evaluation. (…)”
  • Biophysical biomarkers were introduced earlier and directly linked to cancer detection in line 43-45 “In this context, biophysical biomarkers, such as extracellular matrix stiffness and cellular mechanical cues, have gained interest as potential indicators for cancer detection. These biomarkers influence key cellular processes such as differentiation “, responding to both reviewers’ request for a clearer rationale. Redundancies were removed to enhance readability, addressing Reviewer 1’s concern about conciseness.
  1. Microfluidic Systems for Cell Separation and Enrichment (Page 2-3)
  • The previous title “Microfluidic Systems for Cellular Analysis” was changed to “Microfluidic Systems for Cell Separation and Enrichment “and moved from section 5 to 2, aligning with Reviewer 2´s suggestion. As a result, figure 7 as altered to figure 1.
  • Moreover, the introductory text of this section was improved for clarity, addressing Reviewer 1’s request to enhance readability. Line 69 – 83 “Microfluidics involves the manipulation and analysis of small fluid volumes within microscale channels, typically measuring hundreds of micrometres in width. [74] Operating at the submillimetre scale, these systems offer several advantages, including high sensitivity, minimal reagent consumption, cost-effectiveness, and enhanced spatiotemporal resolution. [75] Despite these benefits, challenges remain, particularly in scalability, due to complex device design, prototyping, and fabrication. Other limitations include sample preparation, channel clogging, and surface fouling from biological materials, all of which can compromise reliability and limit further implementation in clinical settings. [76] Nevertheless, the integration of high-throughput microfluidic platforms in oncology is revolutionizing the research field. A notable example is a system capable of generating approximately 12,000 tumour spheroids per chip. [77] This platform allows drug screening in a 3D environment that closely mirrors in vivo tumour conditions, offering a clear improvement over traditional 2D cultures. Importantly, cell viability assessments conducted in a non-destructive, label-free manner agreed with conventional viability assays, demonstrating compatibility with numerous compounds and cell lines. [77] Such innovations hold significant promise, not only for advancing drug development but also for personalized treatment strategies. By offering patient-specific cancer models, these microfluidic platforms have the potential to predict therapeutic responses and assess disease prognosis without the need for labeling or invasive procedures.”

2.1. Passive Cell Separation Microfluidics (Page 3)

  • The section was revised to improve clarity. Line 88-91 “Passive microfluidics refers to the manipulation of flows in micro-structured devices that operate without external forces, such as electric, magnetic, or acoustic fields.[78] By relying solely on the intrinsic hydrodynamic behaviour of particles, passive systems are simpler to operate and generally more reliable. Importantly, passive, label-free microfluidic devices can achieve rapid cell separation with high-throughput compared to active microfluidic systems. [78]”

2.1.1. Deterministic Lateral Displacement (Page 3-4)

  • The section was revised to include the basic principles of passive microfluidics and the rationale for using DLD in cancer detection, line 94-97 “Its principle relies on directing particles through a microchannel array of regularly spaced pillars, which causes particles above a certain critical diameter to be deflected along specific trajectories. “, particularly for isolating tumour cells from blood samples (Reviewer 2). The following reference was added: “This size-dependent sorting makes DLD particularly useful for isolating CTCs and clusters from hematologic components in liquid biopsies, as tumour cells are generally larger than most blood cells. https://pubmed.ncbi.nlm.nih.gov/32949099/”
  • We improved the narrative flow by connecting technical details to clinical applications. “A critical factor in DLD systems´ performance is the pillar design. Poorly optimised pillar shapes can destabilize zigzag trajectories for smaller particles, reducing separation accuracy. To address this, topology-optimised DLD chips have been developed with refined pillar shapes and channel widths[78], achieving miniaturization, improved resistance to clogging and tighter control over the critical diameter​. Validation with polystyrene beads and cancer cells demonstrated better capture purities exceeding 92.5% and recovery rates of 97.1%, outperforming conventional DLD designs“ and synthesized key findings to maintain focus (Reviewer 1).
  • Additionally, a critical evaluation of DLD's limitations was added, such as its reliance on size-based separation, addressing Reviewer 1’s suggestion for deeper analysis of challenges on line 114-116. ”Nonetheless, current DLD platforms remain limited to size-based discrimination, with minimal sensitivity to cellular deformability or shape. Therefore, integrating DLD with complementary active sorting or functional assays would improve specificity and enable functional characterization, further increasing translational value.”

 2.1.2. Inertial Focusing and Centrifugal Microfluidics (Page 4)

  • Regarding consistency, we aligned terminology, such as “separation efficiency” and “capture efficiency,” and standardized device names like CTC-iChip, as suggested by both reviewers. We also enhanced the narrative flow by linking application examples more fluidly, ensuring smoother transitions between topics, as pointed out by Reviewer 1. Additionally, we fine-tuned technical phrases for better scientific precision, in line with the reviewers’ recommendations for greater clarity and coherence. All these minor changes are highlighted in the document in blue as well from lines 119-145.

    2.1.3. Microfiltration (Page 5)

  • In response to the reviewers' feedback, we revised the text to highlight the challenges of isolating rare CTCs, particularly in samples with low concentrations, such as 0.3 CTCs/mL in pancreatic cancer. The revised text from line 163: "Moreover, the microfiltration technique has been applied for the negative enrichment of CTCs from whole blood, selectively depleting normal blood cells while preserving tumour cells.[29] This strategy reduced bias and increased the technology´s applicability across different tumour types. Notably, Chia-Heng Chu et al. [29] developed a 3D-printed microfluidic device that combined leucodepletion channels with microfiltration. It effectively separated tumour cells from blood based on size contrast with normal cells, achieving a 2.34-log depletion by capturing over 99.5% of white blood cells from 10 mL of whole blood while recovering more than 90% of tumour cells.[29] This device successfully isolated CTCs from prostate and pancreatic cancer patient samples, with measured concentrations ranging from 0 to 3.4 CTCs/mL in prostate cancer samples and around 0.3 CTCs/mL in pancreatic cancer samples. .[29] While these concentrations are relatively low, highlighting the challenge of isolating rare CTCs, they still demonstrate the device's significant potential for non-invasive tumour diagnostics and liquid biopsies based on biophysical features of cells." This change emphasizes the difficulty of CTC detection at low concentrations while maintaining the focus on microfiltration's potential for non-invasive diagnostics and personalized medicine. Additionally, we adjusted the sentence in lines 173-178 to improve clarity: " In conclusion, microfiltration is a powerful and versatile tool for cell separation, offering significant improvements in isolation efficiency, throughput, automation, and compatibility with downstream analyses. However, as highlighted, the low concentration of CTCs in certain samples may present challenges in detecting and isolating tumour cells, which warrants further optimization for more sensitive applications. Combining microfiltration with active separation methods further enhances accuracy. Technique’s adaptability to various clinical oncological applications underscores its promise in enhancing diagnostic capabilities and personalized medicine." Also, Figure 1 legend was simplified for clarity.

2.2. Active Cell Separation Microfluidics (Page 7)

  • Figure 8 has been renumbered to Figure 2. We also removed some CTC abbreviations to streamline the text.

2.2.1. Acoustofluidics (Page 7-8)

  • Changes include synthesizing the text for clarity, specifying yield values (25-50% for density gradient centrifugation vs. 82% for microfluidics). Line 215-227 now reads “Traditionally, centrifugation using density gradients is commonly used for isolating blood components, like RBCs, WBCs, and platelets. However, this method has limitations. Density gradient centrifugation yields only 25-50% in efficiency and suffers from low biocompatibility, as it requires additional buffer preparations and can lead to exosome fusion and soluble protein contamination, which compromises the integrity of isolated particles.[32] In contrast, acoustofluidics provides a more efficient (~82% yield) biocompatible alternative by allowing continuous, gentle separation of cells without the need for chemical labels or extensive sample preparation. For example, in plasma exchange applications, acoustofluidics successfully guided RBCs into clean medium channels, achieving >95% recovery and 98% contaminant removal at a flow rate of 0.17 mL/min, while preserving cell viability during platelet and WBCs separation.[32] In oncology, acoustofluidics has proven particularly useful for isolating CTCs from blood- an inherently difficult task due to their low abundance. Using a tilted angle standing surface acoustic wave (SSAW) platform, CTCs were isolated at a flow rate of 1.2 mL/h with a recovery rate above 83%, even when samples contained as few as 100 CTCs/mL.[33] Optimization through finite element modeling of device geometry, tilt angles, and operational parameters ensure efficient cell focusing and separation.[33].”, providing a clearer comparison of the techniques. The examples of biocompatibility issues, like exosome fusion and protein contamination, were more thoroughly explained as well.

2.2.2. Magnetofluidics (Page 8-9)

  • The definition of ferrofluidics was added from lines 260-273 explaining that it “involves using magnetic fields to control and separate ferrofluids based on their magnetic properties”. Separation rates were compared, highlighting the effectiveness of different techniques “Notably, the separation was achieved at an even higher rate than the previously reported 1.2mL/h”.
  • The importance of circulating hybrid cells studies was clarified to emphasize that there are more relevant cell populations beyond conventional CTCs in microfluidics applications. Line 274-282 “Beyond CTCs, circulating hybrid cells (CHCs) have gained attention in recent magnetofluidics studies due to their unique characteristics, combining attributes of both macrophages and neoplastic cells, and their functional role in metastatic spread [40]. These hybrid cells, often more abundant than conventional CTCs in cancer patients [41], have been found to levitate and focus on different heights under paramagnetic conditions, allowing quantification of their biophysical properties. Specifically, CHCs exhibited levitation heights of 320 ± 29 μm in a 30 mM medium, which increased to 422 ± 19 μm and 468 ± 27 μm in higher concentrations of paramagnetic medium of 50nM and 80 mM, respectively, contrasting with the lower levitation heights of peripheral blood mononuclear cells (PBMCs) (average of 226 ± 58 μm in 30 mM, 295 ± 75 μm in 50 mM, and 400 ± 84 μm in 80 mM). [40] These findings highlight the distinct biophysical properties of CHCs, which can be exploited for their detection and isolation among healthy cells.” The reference 41 was added: Relevance of circulating hybrid cells as a non-invasive biomarker for myriad solid tumours, Scientific Reports, 10.1038/s41598-021-93053-7.
  • A comment regarding a limitation of this method was introduced in lines 283-292: “In sum, magnetofluidics is a powerful and versatile platform for cancer cell separation and detection, leveraging intrinsic magnetic susceptibilities to isolate rare cells with high efficiency and minimal labeling. Compared to traditional methods like density-based centrifugation or flow cytometry, magnetofluidics systems offer label-free sorting specificity, particularly in capturing CTCs based on their inherent single-cell density lower levitation heights. Besides, it allows a detailed cell mass biodensity characterization. While magnetofluidics offers promising capabilities, the need for complex and expensive apparatus for micromachining remains a significant obstacle to the rapid and low-cost fabrication of lab-on-chip (LOC) systems based on magnetophoresis. This limitation can hinder the widespread accessibility and scalability of these technologies for broader applications.[42] Nevertheless, advances in ferrofluid formulations, levitation chamber designs, and hybrid technologies will likely elevate its clinical relevance in non-invasive cancer diagnostics and personalized oncology.” The reference 42 was added: Recent advances and current challenges in magnetophoresis based micro magnetofluidics, Biomicrofluidics, 10.1063/1.5035388.

2.2.3. Dielectrophoresis Microfluidics (Page 10-12)

  • In this section, alterations were performed to improve clarity and text integration with figures: Lines 305-325. Moreover, a critical challenge in developing the new electrode configurations was highlighted: inconsistencies in cell capture due to the alignment and positioning of electrodes. This issue is particularly significant when scaling up the technique for high-throughput applications or clinical settings, where precise and reproducible results are vital for successful proof-of-concept studies. “In oncology research, DEP has proven effective for isolating CTCs from leukocytes without compromising cell viability.[45] In a study evaluating different lateral electrode configurations, a semicircular electrode arrangement was found to be the most effective, achieving a recovery rate of nearly 95% for MDA-MB-231 breast cancer cells. The semicircular configuration applied a maximum electric field of 1.11 × 10⁵ V/m, below the threshold required for cell electroporation, ensuring that the cells maintained their integrity during the separation process. Joule heating studies also indicated minimal temperature variation, approximately 1 K, which is a safe range for maintaining cell viability within the microchannel fluid. Overall, the findings confirm the effectiveness of dielectrophoresis in tumour cell separation.[45] In the context of CTC isolation from blood samples, including red blood cells (RBCs), white blood cells (WBCs), and platelets, a microfluidic system incorporating front electrode configuration dielectrophoresis (FEC-DEP) has demonstrated significant promise. (Figure 2 – C). Numerical simulations were employed to optimize system parameters for improved separation efficiency and purity, resulting in an approximately 80% separation rate for CTCs. This highlights the technique’s potential for non-invasive tumour cell isolation from blood samples. However, the alignment and positioning of electrodes can sometimes introduce inconsistencies in cell capture, which can impact reproducibility. These factors can pose challenges in scaling up the technique for high-throughput applications or in clinical settings where consistent results are critical for proof-of-concept.[46] Moreover, DEP has shown promise in the capture and detection of other biomarkers of tumours: small extracellular vesicles (sEVs), particularly found in urine. A specific DEP, known as the acDEP-Exo chip, was developed to detect sEVs with a sensitivity limit of 161 particles/µL, far surpassing conventional methods such as immunoafinnity and ultracentrifugation.[47] In another study using plasma samples from breast cancer patients and healthy volunteers, the combination of biomarkers EpCAM and MUC1 provided high diagnostic accuracy, showcasing DEP’s potential for non-invasive biomarker detection without the need for immunofluorescence methods, which can suffer from contamination and mechanical damage issues.[47]”
  • Additionally, the modifications made to previous the Figure 8 (now Figure 2) and its simplified caption provide clarity.
  • The conclusion (lines 342-349) now emphasizes the ongoing difficulty of fine-tuning parameters for different cell types and sample matrices. “Thus, dielectrophoresis stands out as a precise technique for the manipulation and separation of cancer cells. The use of optimized semicircular and frontal electrode configurations demonstrated high efficacy in the recovery and preservation of tumour cells, ensuring their viability. However, there´s still a challenging need to fine-tune parameters for each specific cell type and sample matrix. Proper calibration and alignment of electrode configuration are essential to minimize issues and ensure reproducible performance. Furthermore, the technique has shown promise in detecting tumour biomarkers by capturing sEVs, which are predominantly found in urine, thereby enhancing non-invasive diagnostic approaches. These advances position dielectrophoresis as a valuable method for cell separation and non-invasive diagnosis of diseases like cancer, contributing significantly to the development of high-precision diagnostics.”

2.3. Cellular Mobility within Microfluidic Arrays (Page 12-13)

  • Figure 9 was altered to Figure 3 enhances the overall flow and structure of the presentation.
  • The inclusion of ovarian cancer cells as a specific example (lines 367-372), “Recent studies revealed that cell migration is influenced by factors such as matrix stiffness, interstitial flow, and interactions between tumour cells and cancer-associated fibroblasts. Furthermore, tumour cells exhibited distinct migratory behaviors, alternating between individual and collective migration, in response to biochemical and biomechanical signals.[49] Real-time visualization of cellular dynamics allows for the observation of these migratory behaviors under different experimental conditions, providing valuable insights into the metastatic process. To illustrate this, a study revealed that metastatic HEY and OVCAR-3 ovarian cancer cells exhibited less deflection when passing through microfluidic ridges compared to IOSE non-metastatic cell lines, which were more strongly deflected[48]. This finding highlights the differences in biomechanical properties that correlate with metastatic potential (Figure 3)”) gives a clearer context and improves the reader's understanding of the experimental setup seen. This update ensures that the key example is directly tied to the visual representation, as asked by Reviewer 1.

  1. Label-Free Microscopy Techniques for Cellular Analysis (Page 14)
    3.1. Phase-Contrast Microscopy: Morphological Profiling and Segmentation (Page 14-16)
  • Restructuring of Section 2 to Section 3, along with the reordering of Figure 1 to Figure 4. Moreover, the text was altered for improvement.
  • An explanation of the method was added from lines 392-395 for methodological clarity: “This method exploits the phase shift of light waves passing through the sample, converting these shifts into visible contrast to enhance the image.[51] By amplifying interference patterns, phase-contrast microscopy provides high-contrast images that reveal fine details of the sample, while examining living cells in their natural state, without staining.[52]”
  • Its conclusion now effectively ties back to the disadvantages discussed throughout the text (lines 446-452), offering a reflective summary. It now reads”Phase-contrast microscopy has significantly evolved, particularly when combined with complementary techniques, and has proven to be a highly effective method for the detailed observation and analysis of cancer cells. However, challenges remain, including the limitations in quantifying optical thickness, difficulties in segmenting densely packed or overlapping cells, and the potential for segmentation errors in complex environments. Additionally, the need for significant computational resources and manual validation of automated results can be time-consuming. Despite these challenges, the combination of phase-contrast microscopy with advanced analytical approaches continues to offer promising applications for cancer research and beyond.“ This alteration ensures that the key challenges highlighted earlier are properly addressed in the concluding remarks, as peer reviewers 1 and 2 suggestions.

3.2. Holographic Microscopy: Quantitative Phase Imaging for 3D Characterization (Page 16-18)

  • QPM abbreviation for Quantitative Phase Microscopy, was added (line 455).
  • Reorganization of Figure 2 into Figure 5, along with the simplification of its caption, streamlines the visual presentation, making the figure more readable.
  • Several paragraphs were rephrased, resulting in a more concise and coherent flow of information.
  • Furthermore, the disadvantages discussed earlier were included in the conclusion (lines 533-537) “In summary, holographic microscopy is an innovative, non-invasive technique for 3D analysis of living cells, which is advantageous for studying tumour dynamic processes and cellular morphology Its ability to quantify optical phase and cell morphology in real-time, along with its compatibility with other techniques, makes it indispensable in cancer research and beyond. However, challenges such as pixel size limitations, high sampling size, and the presence of noise must be addressed to improve its resolution and applicability.”

  1. Cytometric Techniques (Page 20)
  • The introductory text was altered for clarity from line 563: ”Cytometry encompasses a range of analytical methods designed to quantify the physical and chemical characteristics of individual cells or particles. Traditionally, these methods have relied on light scattering and fluorescence emission to evaluate parameters such as size, granularity, and biomarker expression[45]. Modern cytometric techniques have introduced label-free strategies that expand the scope of cellular features analyzed, such as electrical impedance, mechanical deformability, and morphology. In label-free cytometry, cells typically traverse laser beams or electrical fields, with detectors capturing real-time data on their intrinsic properties.[79] The following subsections explore several of these modalities that have shown promise in cancer research and diagnostics”

4.1. Deformability Cytometry: Mechanical Fingerprinting of Cells (Page 20-21)

  • The addition of the early deformability cytometry comparison (lines 572-591) sets the stage for clearer contrasts between different modalities: Deformability cytometry allows the analysis of cells' mechanical properties, such as deformability and viscoelasticity - features closely linked to key biological processes, such as cell differentiation, therapeutic response, and disease progression.[80] Real-time assessment of these biophysical properties provides valuable insights into dynamic cell behaviors under physiological and pathological states. Early deformability cytometry platforms used extensional flow to stretch cells, achieving throughputs of up to 1,000 cells per second. Deformability was quantified by the maximum aspect ratio during forced deformation, offering greater throughput and sensitivity than previous techniques for studying cell deformation under stress.[80] Notably, one of the first applications of this approach involved using Latrunculin B to treat HL60 leukemia cells, revealing increased deformability after acting disruption. These mechanophenotypic changes distinguish treated and untreated cancer cells, highlighting the sensitivity of deformability cytometry to subtle cytoskeletal alterations, which other cytometric techniques might not detect.[81] A more recent development, viscoelastic deformability cytometry (vDC), was introduced by Asghari et al.[82] and improved upon earlier platforms by offering a high-throughput, label-free platform capable of analyzing up to 100,000 cells per second. This is achieved through parallelized microfluidics and viscoelastic flow (Figure 6-A) to perform real-time mechanical phenotyping of both liquid and solid biopsies. By assessing cell deformability and size, vDC can distinguish malignant from normal cells. For example, RBCs and PBMCs differed in deformability, while breast cancer cell lines were distinguished by size. Importantly, vDC demonstrated its diagnostic potential by identifying chronic lymphocytic leukemia (CLL) cells, which exhibited significantly higher deformability than healthy B cells or PBMCs. This approach was also able to differentiate glioma-initiating cells (GICs) from more differentiated glioma cells and astrocytes, and it successfully detected rare cancer cells in blood, making it particularly useful for liquid biopsy applications.[82] These results established vDC as a robust, label-free method for cancer detection and disease monitoring via mechanical phenotyping.” The text highlights the improvements of Viscoelastic Deformability Cytometry (vDC), which increases throughput to 100,000 cells per second
  • Meanwhile, Constriction-based Deformability Cytometry (cDC), although slower at 25,000 cells per minute, captures more biomechanical parameters, as mentioned in line 594: “Although the rate is slower than earlier platforms (Figure 6 – B). cDC captures a broader range of biomechanical parameters, including passage time, area-in-constriction, and deformation index.” Also, stratification accuracy was noted from line 600: “Notably, it classified metastatic phenotypes with over 92% accuracy using a support vector machine model trained on multiparametric data and achieved 89.5% accuracy in distinguishing cancer cells from leukocytes in a blood-like context. The platform also incorporated a physics-based viscoelastic model grounded in power-law rheology, enabling stiffness estimation that aligned with atomic force microscopy benchmarks. In contrast, earlier systems were unable to achieve this level of stratification.” These revisions clarify the strengths of each modality and compare them, as asked by both reviewers.

4.2. Impedance Cytometry: Electrical Cell Characterization (Page 22-24)

  • Impedance cytometry is now clarified as a standout method compared to other cytometries, especially due to its ability to characterize cells without the complexities associated with optical techniques. As stated in lines 624-626: This cytometry stands out for its ability to characterize cells without the complexities of optical methods, eliminating the need for high-transparency devices or frequent calibration of laser beams.”
  • While impedance cytometry has the significant advantage of distinguishing live from dead cells, the MIC device operates at a slower rate compared to other systems. As highlighted in lines 891-893: “For example, a microfluidic impedance cytometry (MIC) device, incorporating a novel double-differential electrode configuration, demonstrated a 94.5% accuracy in classifying live versus dead MCF-7 breast cancer cells, although only at a rate of approximately 1,000 cells per second.” Despite its slower throughput, this high accuracy is a notable strength.
  • A potential disadvantage of the AI-based model used in impedance cytometry has been addressed in lines 664-668: However, a potential disadvantage of this approach is that the accuracy and structural fidelity of the AI-based model could heavily rely on training data quality. If the model is not trained with a sufficiently diverse set of cell types or conditions, it may struggle to generalize or misclassify cells outside the trained dataset. Moreover, the system might face challenges with highly heterogeneous or rare cell populations, such as CAFs or CTCs, where impedance signals alone may not provide enough distinct features for reliable classification.[88]“ Reference 88 was added: A review on intelligent impedance cytometry systems: Development, applications and advances, Analytica Chimica Acta, 10.1016/j.aca.2023.341424

4.3. Imaging Flow Cytometry: High-Throughput Morphological and Functional Analysis (Page 25)

  • Changes in text flow and improvements to the overall coherence and readability were performed. Line 668-698 “Imaging flow cytometry (IFC) combines the capabilities of traditional flow cytometry with the detailed morphological analysis of microscopy. By capturing high-resolution images at high-throughput, IFC enables single-cell analysis with precise morphometric cellular analysis, making it a powerful tool for label-free cancer cells detection.[89] Unlike conventional flow cytometry, which relies on fluorescent probes, label-free IFC eliminates issues like non-specific probe binding, interference and low quantitative performance. It overcomes the inherent challenges of labeling, such as variability in probe binding and cellular molecule labeling, which can compromise detection accuracy.[90] Despite its advantages, label-free IFC is not without challenges. These include high data complexity, the need for advanced instrumentation, and difficulties with data storage and analysis. Additionally, the high-throughput nature of IFC can complicate equipment handling, and software limitations, such as manual region definition, still exist.[91] To address them, recent innovations in the field have focused on integrating complementary imaging modalities and AI-driven analysis to streamline data interpretation and enhance the precision of cell classification.”

4.3.1. Multimodal Integration and AI-Enhanced Applications in IFC (Page 25-27)

  • Rewriting of the IFC example for detecting bladder cancer cells (from line 712) simplified the explanation, making it more accessible while retaining key details as asked from the reviewers. The revised text now reads: Another promising application of IFC is its use in detecting bladder cancer cells in urine. A quantitative interferometric IFC (QIFC) system developed by Dudaie et al.[94] employed interferometric phase microscopy (IPM) to capture optical path delay (OPD) maps of individual cells as they flow through the system (Figure 8-A). By segmenting cells using deep learning and machine learning algorithms, the system extracted 20 biophysical features, including morphology and OPD-related parameters such as dry mass and optical volume, to classify cells as benign or malignant. This platform achieved 96% accuracy and 96% AUC in clinical urine samples, outperforming some FDA-approved urinary biomarkers.[94] It demonstrated significant differences in cell shape, internal complexity, and dry mass between healthy urothelial cells and bladder cancer cells. Malignant cells had higher circularity and mean OPD, but lower area, OPD energy, OPD volume, dry mass, OPD entropy, and contrast compared to healthy cells (all p < 0.001). Similarly, deep learning-enhanced image cytometry (DLIC) has shown great promise in detecting altered PBMCs from extra nodal NK/T-cell lymphoma (ENKTL) patients[95]. By extracting 3 biophysical markers—cell size, eccentricity, and refractive index (RI), from over 270,000 PBMCs, DLIC revealed stage-specific phenotypic heterogeneity that correlated with relapse and treatment status. (Figure 8 – B). Mean size increased (80.9 μm² to 81.9 μm²), eccentricity decreased (0.59 to 0.56), and RI slightly decreased (1.3915 to 1.3911) from the interim to end-of-treatment groups[95]. Their system achieved on-site, label-free quantification, laying the foundation for scalable, precision monitoring in hematologic malignancies for clinical decision support.”
  • In the conclusion, challenges were incorporated to reinforce the need to address them for successful AI-IFC integration. Line 751-755: With these recent advancements, the integration of IFC with AI and other microscopy techniques has made IFC a promising technology for label-free cancer diagnosis. It stands out as a high-throughput cellular analysis with high-resolution imaging, surpassing limitations of conventional flow cytometry. However, integrating AI with IFC presents challenges, such as accurately interpreting complex data, handling short time-point phases, and ensuring consistent performance. Addressing these challenges will be key to fully unlocking the potential of IFC for oncology.” This conclusion ties together the technical and practical hurdles AI faces in improving IFC-based diagnostics.

  1. Cell and Particle Scattering Techniques Page (27)

·      An introduction was added to improve clarity, text flow and integration with the subsections as requested by the reviewers. Line 758 “Scattering-based biophysical techniques have emerged as powerful, label-free approaches for probing cancer cell properties and their surrounding microenvironment. By analyzing the interaction of light with cells, extracellular vesicles, and biomolecules, these methods provide valuable information on particle size, surface charge, and molecular composition, all of which are closely linked to tumour progression and metastatic potential. Unlike conventional biochemical assays, scattering techniques enable real-time and non-destructive measurements, often with minimal sample preparation, thus making them highly attractive for translational cancer research and diagnostics. The next section will address electro-mechanical and surface characterization techniques, which further extend our ability to probe cancer cell structure and function at the nanoscale.”

5.1. Dynamic Light Scattering and Zeta Potential (Page 27-29)

  • A distinction between DLS and SLS resolution (line 770) was added to enhance the key methodological differences as requested: Dynamic light scattering (DLS) is a label-free technique originally designed for characterizing particle sizes in suspension, particularly nanoscale particles and polymers. It relies on measuring temporal fluctuations in the intensity of scattered light, providing information about particle size distributions in a sample.[97] In contrast, static light scattering (SLS) measures the angular intensity distribution of the scattered light for larger particles or aggregates.” This helps differentiate the two techniques clearly and concisely. Additionally, 774 now highlights DLS's relevance in analyzing exosomes: Notably, DLS is well-suited for exosomes analysis, nanoscale extracellular vesicles implicated in cancer biology due to their role in cell-to-cell communication.” This emphasizes the specific application of DLS in cancer research, improving the text's focus.
  • DLS limitations (lines 777) were simplified for clarity: However, DLS presents some limitations. It is sensitive to environmental factors, such as temperature fluctuations and solvent viscosity, which can influence the data quality. Other challenges include low particle concentrations, signal-to-noise ratio issues, and artifacts like bubbles or reflective particles that reduce accuracy. These factors would need careful experimental control, larger sample volumes, and samples with higher RI to ensure reliable results.[100] In cancer research, DLS provides insights into cellular dynamics, such as aggregation tendencies and changes in cellular motility that correlate with cancer progression. Studies have shown how the zeta potential (surface charge) of cancer cells measured varies (Figure 9-A), with the pH and ionic strength of the surrounding medium. This revised version condenses the limitations while retaining all critical information.
  • Conclusion has been simplified for better readability: Overall, DLS has been showing promising applications as a non-invasive technique for cancer diagnosis, with relevance in cancer cells and extracellular vesicles analysis. Nonetheless, its current use in routine clinical practice remains limited, primarily due to challenges with low signal sensitivity and the need for high-quality samples. The integration with complementary techniques could make it a powerful tool for identifying biomarkers associated with cancer, particularly through the study of extracellular vesicles. Its capabilities make it a useful technique for investigating cellular interactions, with significant potential for identifying biomarkers associated with cancer.”

5.2. Surface-Enhanced Raman Spectroscopy (Page 29-30)

  • Limitations of the technique were added: “However, this technique has limitations, including the complexity involved in producing reliable plasmonic substrates and the need for more robust nanoparticles to ensure reproducibility. Nonetheless, ongoing advancements in nanoparticle design and substrate development are continuously improving the performance and reliability of SERS for clinical applications.[107]”
  • The clarification of AgNP use in SERS technology for kidney cancer now explicitly connects the technique to relevant cancer-specific biomarkers: “More specifically, research into the effects of varying silver nanoparticle (AgNP) concentrations demonstrated significant spectral changes directly related to kidney disease/cancer-specific biomarkers, notably hypoxanthine and uric acid.” This addition strengthens the relevance of SERS in kidney cancer diagnostics.

  • The description of the new SERS system coupled with AI (lines 846-852) was simplified to emphasize key advantages, particularly the small sample volume: A study demonstrated that using only 15 µL of serum with silver nanowires to enhance the SERS signal provided a major advantage by reducing sample requirements, making the method less invasive and more suitable for clinical use. The resulting spectra were analyzed with a support vector machine (SVM) model across 50 Raman intervals (600–1800 cm⁻¹), achieving excellent diagnostic performance with an accuracy of 95.81%, sensitivity of 95.40%, and specificity of 95.87%. The study included over 1,900 samples across five cancer types, namely lung, colorectal, hepatic, gastric, and esophageal, enhancing model robustness. This method also distinguished early cancer from other diseases and showed potential in identifying subtle spectral differences between cancerous and non-cancerous samples. These results highlight the feasibility of applying SERS-AI systems as a tool capable of adapting to different cancer types or patient needs.” The alterations highlighted the clinical applicability of the system straightforwardly, underscoring its minimal invasiveness and high performance.
  • Conclusion was altered to include a smooth transition to the next section on AFM and EIS, introducing them as complementary techniques (lines 857-863): In conclusion, SERS has remarkable potential in cancer diagnostics due to its high sensitivity and specificity. Its application in liquid biopsies and integration with AI systems provides a robust biophysical profile of cancer cells. Despite ongoing challenges related to substrate robustness and particle consistency, ERS is poised to become an indispensable tool in clinical oncology, particularly when combined with microfluidic technologies for single-cell analysis. The next section will cover atomic force microscopy (AFM) and electrical impedance spectroscopy (EIS). AFM and EIS are key techniques that complement scattering methods by providing detailed insights into cell structure and conductivity, essential for understanding all cancer cell biophysical features.” This sets the stage for the upcoming discussion, linking the current content to the next focus on other critical biophysical analysis methods.

  1. Electro-Mechanical and Surface Characterization (Page 30)
  • Section 6 title was altered to “Electro-Mechanical and Surface Characterization” to introduce the content.
  • An introduction was added for clarity and contextualization“In addition to scattering methods, techniques that probe the mechanical and electrical properties of cells provide complementary insights into cancer biology. These approaches capture how cancer alters cell stiffness, elasticity, adhesion, and membrane conductivity, features closely linked to tumour aggressiveness and therapy response. Atomic Force Microscopy (AFM) enables nanoscale assessment of cell mechanics and surface structure, while Electrical Impedance Spectroscopy (EIS) evaluates membrane capacitance and resistance to monitor changes in cell physiology. Together, these methods expand our understanding of tumour biomechanics and electrochemical signatures, offering potential for early diagnosis, prognosis, and personalized treatment strategies.”

6.1. Atomic Force Microscopy (Page 30-32)

  • The description of how AFM operates (lines 875-883) was detailed, clarifying its function: AFM is a versatile technique that provides characterization, manipulation and exploration of cellular surfaces with molecular precision at the nanoscale, as well as the dynamics of membranes. AFM operates by scanning the cell surface with a sharp probe, enabling the measurement of both structural organization and mechanical properties, such as adhesion and stiffness, under near-physiological conditions.[110] This minimal sample preparation, typically limited to fixation to the surface, allows AFM to study cells without extensive modification, preserving the integrity of biological features. Moreover, AFM’s sensitivity to vertical displacements allows for precise measurements of cellular mechanics, such as the elasticity and deformability of individual cells.[111] As for limitations, it has low throughput, a restricted imaging area, and relatively low spatial resolution compared to other microscopy techniques. These challenges may limit its application in high-throughput screening or large-scale analyses.[112]
  • The examples of AFM measurements in ovarian and lymphoma cancer cells (lines 1594-1600) were simplified for readability: Over the last years, AFM has been invaluable for characterizing the mechanical properties of cancer cells (Figure 10). Specifically, malignant breast cancer cells were found to have altered mechanical characteristics such as increased deformability and reduced stiffness compared to non-malignant cells [112,113]. Similarly, AFM measurements of Young’s modulus of ovarian cells reveal that non-malignant ovarian IOSE cells had the lowest stiffness (2.47 ± 2.05 kPa), followed by HEY cells (0.88 ± 0.53 kPa), and the highest stiffness in HEY A8 cancer cells (0.49 ± 0.22 kPa), indicating a correlation with metastatic potential. Raji lymphoma cancer cells had lower stiffness (0.2 to 0.4 kPa) compared to non-cancerous T-lymphocytes (1 to 1.4 kPa) and K-562 bone marrow cells (0.6 to 0.7 kPa), suggesting that more aggressive cells are more deformable.[114].” The text now conveys the same essential data for clarity.
  • Similarly, other examples were also simplified (lines 892-899): The deformability of cancer cells, assessed by AFM, is also significantly higher than that of non-malignant cells (Figure 10 – A), which served as a biomarker of bladder cancer cells.[117] Healthy bladder cells (HCV29) had a higher modulus of elasticity, ranging from 50 kPa at 100 nm to 10 kPa at 500 nm, compared to cancerous cells (HTB-9), which ranged from 15 kPa at 100 nm to 5 kPa at 500 nm, confirming that healthy cells are stiffer than cancer cells.[117] This was also verified in breast cancer cell lines, due to well-organized actin fibers(Figure 10 – B). At 1 Hz, cell stiffness decreased with malignancy: MCF-10A (0.7 ± 0.3 kPa), MCF-7 (0.5 ± 0.1 kPa), and MDA-MB-231 (0.3 ± 0.1 kPa). At 250 Hz, all cells became stiffer, but the stiffness difference between healthy and cancer cells increased: MCF-10A (250 ± 100 kPa), MCF-7 (28 ± 12 kPa), and MDA-MB-231 (25 ± 13 kPa).[118] This version offers a clearer contrast between the stiffness of different cell lines at varying frequencies.
  • Conclusion was rewritten and reduced for conciseness: Overall, AFM’s ability to probe nanoscale mechanical properties provides essential insights into tumour biomechanics, offering potential for early cancer diagnosis and therapeutic assessment, though its application remains limited by technical constraints such as low throughpu” The revision strengthens the statement by focusing on the main disadvantage, which is the low throughput of the technique.

6.2. Electrical Impedance Spectroscopy (Page 33-34)

  • EIS technological basis was added for clarity, as well as, some disadvantages from line 932: “Electrical impedance spectroscopy (EIS) is a technique that has been applied for the early detection of neoplasms.[111] It measures the resistance and capacitance of cell membranes at varying frequencies, providing insight into the cell’s internal structure, size, and viability.[111] Unlike Zeta Potential measurements, which focus on surface charge, EIS assesses the overall impedance of the cell, offering a broader picture of its physiological state. However, this technique faces challenges in microfluidic devices due to the need for high sensitivity in detecting signals in small volume sensors[120] Additionally, the large volume of data generated, and potential interference require advanced statistical and computational methods for proper analysis.[121]”
  • The original complex passage from lines 939 was streamlined as follows: Nonetheless, EIS has proven effective in spotting cancer by comparing the electrical properties of cancerous and healthy cells.[111] Comparing bioimpedance tissue measurements in breast cancer patients, the results showed lower impedance values for cancerous tissues compared to healthy ones. Its sensitivity was 77.8%, much higher than traditional breast cancer methods like mammography, which has a false negative rate of 4% to 34%. Cole-Cole parameters, which are used in EIS to represent the frequency-dependent dielectric properties (permittivity and conductivity). It was observed that in seven out of nine patients, cancerous tissues consistently exhibited lower internal resistance and membrane capacitance compared to healthy tissues.[111]” This version simplifies the language without losing the important comparison to mammography’s sensitivity and false-negative rate.
  • Similarly, clarity was improved in the following paragraphs: At the single-cell level, EIS has also identified cancer cells by distinguishing between various stages of cancer progression.[122] EIS analysis of individual cells showed that normal breast MCF-10A cells had a membrane capacitance of 1.94 ± 0.14 AF/cm² and resistance of 24.8 ± 1.05 MΩ at 100 kHz. Meanwhile, increasing metastatic cancer cell lines showed progressively lower capacitance: 1.86 ± 0.11 for MCF-7, 1.63 ± 0.17 for MDA-MB-231, and 1.57 ± 0.12 for MDA-MB-435. Compared to healthy cells, membrane capacitance decreased by 4.1%, 16.0%, and 19.1%, respectively. Electrical resistance remained similar, from 24.8 ± 0.93 MΩ for MCF-7 to 26.2 ± 1.07 MΩ for MDA-MB-435.[122] Nevertheless, the decrease in capacitance suggests that as cancer progresses, the cell membrane becomes more permeable, which is an EIS measurement that could be used as an indicator of disease stage. EIS has also shown value in identifying metastatic potential and monitoring chemoresistance, the ability of cancer cells to resist chemotherapy, by detecting resistant cells before symptoms manifest.[123,124] Less metastatic cells exhibited a significantly higher impedance phase value of -63.4° ± 8.6°, while highly metastatic cells showed a lower value of -73.4° ± 10.4° (P < 0.001).[123] The technique also monitored drug-induced cell adhesion, proliferation, and death, revealing a decrease in impedance as cells became chemoresistant, indicating changes in their electrical properties. Additionally, EIS facilitated the identification of cellular plasticity. This early detection is crucial for personalized cancer treatment strategies, allowing for timely intervention.[124] This revision clearly communicates the technical findings while emphasizing the implications for cancer progression.
  • Conclusion was rewritten for conciseness (line 962-967):
    In conclusion, EIS is a powerful tool for cancer detection by monitoring changes in electrical properties in cancer cells. Overcoming the challenges of sensitivity in small-volume sensors and improving accuracy will be crucial for expanding the clinical use of EIS. Then, features, such as capacitance and resistance, can indicate cancer progression, metastatic potential, and chemoresistance. EIS also allows for the monitoring of cellular plasticity, making it possible to identify drug-resistant cells before clinical symptoms appear. These capabilities highlight EIS’s potential for early diagnosis and personalized treatment strategies in cancer management.”

  1. Clinical Translation (Page 34-38)
  • We revised the clinical translation section to incorporate a more integrated critical appraisal across all techniques, addressing challenges and opportunities for each method throughout the review rather than” focusing primarily on this aspect at the end”. Table 1 was added to reflect those (please see next page). Figure 11 legend was improved for clarity.
  1. Conclusions and future perspectives (Page 38)
  • Reviewer 1's feedback was taken into consideration, particularly by addressing current barriers to clinical adoption and providing a more balanced view of the limitations of the technologies. The structure was streamlined for easier reading, as per Reviewer 2's feedback on improving organization. Furthermore, the future research direction emphasizes the importance of collaboration and highlights the need for standardized protocols and longitudinal studies, responding to the call for a more comprehensive view of clinical translation. Lines 1024-1043 “Label-free technologies that assess biophysical cellular properties have emerged as powerful tools in cancer research and diagnostics. They offer a unique perspective into cell behavior without the need for molecular labeling or extensive samples pre-processing. By leveraging intrinsic biophysical cues, such as cell size, shape, stiffness, dielectric properties, and surface adhesion, these techniques enable real-time, non-invasive characterization of cancer cells vs normal cells, including rare populations such CTCs. As highlighted in this review, several platforms, ranging from optical imaging (e.g., holographic and phase-contrast microscopy), electro-mechanical profiling (e.g., impedance cytometry, atomic force microscopy), and microfluidic technologies (both passive and active), have demonstrated substantial potential in detecting, isolating, and phenotyping cancerous cells. Despite their promising applications, clinical adoption of label-free biophysical methods faces several challenges. Among these are the lack of standardized protocols, limited multicentre validation studies, and technical issues related to reproducibility and device integration. Variability in measurement outputs across different platforms and sample types makes it difficult to establish consistent universally accepted diagnostic thresholds. Furthermore, while AI-powered data analysis offers tremendous potential, its clinical deployment requires transparent, interpretable models supported by rigorous regulatory frameworks to ensure trust and accountability in medical settings. Looking ahead, future research should prioritize the development of robust, hybrid platforms that combine multiple biophysical modalities to increase diagnostic specificity and sensitivity. Advancing towards compact, user-friendly and point-of-care systems will also be crucial to translating these technologies into routine clinical practice. Besides, more longitudinal studies with large patient cohorts are needed to validate the prognostic and predictive value of biophysical biomarkers over time. Nevertheless, collaborative efforts between engineers, biologists, clinicians, and data scientists are essential to translate technological innovations into clinical utility.”

Table 1: Overview of biophysical label-free techniques, their challenges and research gaps for clinical translation.

Biophysical

Label-free method

Technique

Challenges

Requirements for clinical translation

Microfluidic Systems for Cell Separation and EnrichMEMT

Passive Cell Separation Microfluidics

DLD: Limited to size-dependent discrimination; Not sensitive to cellular shape or deformability.

IF: Does not separate cells with similar biophysical features.

MF: Shear stress damage; Not indicated for small volumes of samples

·       Miniaturization of DLD and IF chips.

·       Integration with active sorting or functional assays.

·       Improve resistance of chips to clogging.

·       Enhance capture of cells within mixed populations and similar features.

·       Improve detection for small volumes.

Active Cell Separation Microfluidics

AC: Cell misalignment upon entry; potential for clogging; loss of rare cells post-separation processing

MG: More expensive micromachining

DC: Low single-cell resolution

·       Combine AC with DC

·       Improve pre-alignment strategies.

·       Modelling device geometry, tilt angles and change operational parameters

·       Reduce fabrication costs of chips based on MG

·       Fine-tune parameters for each cell type and sample matrix

Cellular Mobility within Microfluidic Arrays

EICS: Maintaining high cell viability; Only for metastatic potential.

·       Study cell migration across cancer stages and types.

·       Target other features, besides cell behaviour;

MICROSCOPY FOR Cellular Analysis

Phase-Contrast Microscopy

Limited to 2D imaging; Does not quantify subtle changes in depth; Segmentation errors in overlapping cells.

·       Create algorithms for automated validation.

·       Refine segmentation in heterogeneous populations:

·       Eliminate the need for time-consuming manual validation of results.

·       Improved quantification of optical thickness. 

Holographic Microscopy

Requires high computational power for 3D reconstruction; sensitivity to noise

·       Address limitation in pixel size.

·       Minimize the need of large sampling sizes.

·       Enhance the ability of real-time monitoring of OPD and cell morphology.

Cytometry

Deformability Cytometry

vDC: Only assess cell deformability depending on size.

cDC: Lower throughput than vDC;

·       Increase throughput rate of cDC close to vDC rates.

·       Turn vDC less dependent on particle size to stratify more cancer phenotypes.

Impedance Cytometry

Sensitivity issues with small volumes and rare populations; Only relies on impedance feature

·       Increase the quality and diversity of training data for AI-based integration.

·       Comparative studies of accuracy with traditional markers.

·       Multimodal analysis with other features

Imaging Flow Cytometry

High cost and complexity of systems; Requires expert handling of multimodal data; software limitations:

·       Integrate with AI to streamline data interpretation.

·       Eliminate the need for manual region definition;

Cell and Particle Scattering analysis

Dynamic Light Scattering

Sensitive to temperature, pH …; Difficulty in differentiating between similar particles; requires high-quality and volume samples:

·       Combine it with glycocalyx analysis or confocal fluorescent imaging.

·       Study the precise impact of any change on experimental conditions

Surface-Enhanced Raman Spectroscopy

Requires high signal enhancement; Sample damage with intense laser use.; Low throughput.

·       Integrate with microfluidic technologies to increase throughput.

·       Optimize formulation of plasmonic substrates and nanoparticles;

Electro-Mechanical and Surface Characterization

Atomic Force Microscopy

Slow data acquisition time; Difficulty in analysing complex samples

·       Increase imaging area and spatial resolution;

Electrical Impedance Spectroscopy

Sensitivity challenges in microfluidic devices with small sample volumes; Need of advanced statistical modules.

·       Increase sensitivity of impedance in small-volume sensors for scalable applications.

Abbreviations: DLD, Deterministic lateral displacement; IF, Inertial Focusing; MF, Microfiltration; AC, Acoustofluidics; MG, Magnetofluidics; DC: Dielectrophoresis; EICS, Electrical cell-substrate impedance sensing; vDC, Viscoelastic deformability cytometry; cDC, Constriction-based deformability cytometry.

Reviewer 2 Report

Comments and Suggestions for Authors

Calejo et al. provided a nice review on label-free techniques and platforms for detecting cancer in this manuscript titled ‘Label-free cancer detection methods based on biophysical cell phenotypes’. They started by describing microscopy techniques including phase-contrast microscopy (combined with segmentation and morphological clustering) and holographic microscopy alone or in combination with other modalities (e.g. fluorescence). They next covered cytometric techniques, including deformability cytometry and impedance cytometry for characterizing the mechanical and electrical properties of the cells, and multimodal imaging flow cytometry that captures multiple types of imaging / optical signals from cells in solution. In the following section, they discussed cell and particle scattering techniques, specifically dynamic light scattering and zeta-potential (as commonly done on nanoparticles). The same section also covered surface enhanced Raman spectroscopy. The authors dedicated the next section to a detailed summary of various microfluidic platforms for cell separation and enrichment, which is essential for finding rare cancer cells in complex liquid biopsy samples. They went on to summarize recent advances in using atomic force microscopy and electrical impedance spectroscopy to characterize cell surface topology and conductivity, respectively, with potential utilities in cancer detection. Lastly, the authors brought attention to the considerations for clinical translation before a brief conclusion and outlook.

Overall, this is a nice and comprehensive summary of recent advances in label-free cancer cell detection. Literature citations appear to be appropriate and up to date. Before publication, I suggest that the authors consider the following revisions.

First, the microfluidics section seems to be somewhat out of place. Potentially this section could become the first one since separation and enrichment would precede the characterizations?

Second, different techniques and platforms are designed to work with different types of samples, such as circulating tumor cells or tumor tissues. It may be a good idea to organize the flow from cells (which would go well with the microfluidics section being the 1st) to tissues or biopsies.

Lastly, for each technical section, before getting into the examples from published reports, it would be helpful to briefly introduce (1) the basic principles of the technique and (2) the rationale for using the technique for cancer detection – that is, why it should work or the biophysical basis of the detection strategy.

Author Response

(The authors gave the same response as above.)

Round 2

Reviewer 1 Report

Comments and Suggestions for Authors

The revised manuscript has undergone substantial improvements. The authors carefully addressed all reviewer comments, reorganizing sections for better flow (e.g., moving microfluidics upfront), simplifying figure captions, and adding Table 1 to summarize translational challenges. The narrative is now clearer, with stronger synthesis and critical evaluation of each technique’s strengths and limitations.

The manuscript is comprehensive, well-structured, and clinically relevant, covering a wide range of label-free methods while highlighting barriers to clinical adoption. Only minor editorial adjustments remain, such as streamlining overlap between the “Clinical Translation” and “Conclusions” sections, polishing a few dense figure captions, and ensuring reference formatting consistency.

I recommend acceptance with minor editorial revisions.